# Cutting Edge of the Pathogenesis of Atopic Dermatitis: Sphingomyelin Deacylase, the Enzyme Involved in Its Ceramide Deficiency, Plays a Pivotal Role

**DOI:** 10.3390/ijms22041613

**Published:** 2021-02-05

**Authors:** Genji Imokawa

**Affiliations:** Center for Bioscience Research & Education and Utsunomiya University, 350 Mine Utsunomiya, Tochigi 321-8505, Japan; imokawag@cc.utsunomiya-u.ac.jp or imokawag@dream.ocn.ne.jp; Tel.: +81-28-649-5229

**Keywords:** atopic dermatitis, pathogenesis, sphingomyelin deacylase, barrier function, ceramides

## Abstract

Atopic dermatitis (AD) is characterized clinically by severe dry skin and functionally by both a cutaneous barrier disruption and an impaired water-holding capacity in the stratum corneum (SC) even in the nonlesional skin. The combination of the disrupted barrier and water-holding functions in nonlesional skin is closely linked to the disease severity of AD, which suggests that the barrier abnormality as well as the water deficiency are elicited as a result of the induced dermatitis and subsequently trigger the recurrence of dermatitis. These functional abnormalities of the SC are mainly attributable to significantly decreased levels of total ceramides and the altered ceramide profile in the SC. Clinical studies using a synthetic pseudo-ceramide (pCer) that can function as a natural ceramide have indicated the superior clinical efficacy of pCer and, more importantly, have shown that the ceramide deficiency rather than changes in the ceramide profile in the SC of AD patients plays a central role in the pathogenesis of AD. Clinical studies of infants with AD have shown that the barrier disruption due to the ceramide deficiency is not inherent and is essentially dependent on postinflammatory events in those infants. Consistently, the recovery of trans-epidermal water loss after tape-stripping occurs at a significantly slower rate only at 1 day post-tape-stripping in AD skin compared with healthy control (HC) skin. This resembles the recovery pattern observed in Niemann–Pick disease, which is caused by an acid sphingomyelinase (aSMase) deficiency. Further, comparison of ceramide levels in the SC between before and after tape-stripping revealed that whereas ceramide levels in HC skin are significantly upregulated at 4 days post-tape-stripping, their ceramide levels remain substantially unchanged at 4 days post-tape-stripping. Taken together, the sum of these findings strongly suggests that an impaired homeostasis of a ceramide-generating process may be associated with these abnormalities. We have discovered a novel enzyme, sphingomyelin (SM) deacylase, which cleaves the N-acyl linkage of SM and glucosylceramide (GCer). The activity of SM deacylase is significantly increased in AD lesional epidermis as well as in the involved and uninvolved SC of AD skin, but not in the skin of patients with contact dermatitis or chronic eczema, compared with HC skin. SM deacylase competes with aSMase and β-glucocerebrosidase (BGCase) to hydrolyze their common substrates, SM and GCer, to yield their lysoforms sphingosylphosphorylcholine (SPC) and glucosylsphingosine (GSP), respectively, instead of ceramide. Consistently, those reaction products (SPC and GSP) accumulate to a greater extent in the involved and uninvolved SC of AD skin compared with chronic eczema or contact dermatitis skin as well as HC skin. Successive chromatographies were used to purify SM deacylase to homogeneity with a single band of ≈43 kDa and with an enrichment of >14,000-fold. Analysis of a protein spot with SM deacylase activity separated by 2D-SDS-PAGE using MALDI-TOF MS/MS allowed its amino acid sequence to be determined and to identify it as the β-subunit of acid ceramidase (aCDase), an enzyme consisting of α- and β-subunits linked by amino-bonds and a single S-S bond. Western blotting of samples treated with 2-mercaptoethanol revealed that whereas recombinant human aCDase was recognized by antibodies to the α-subunit at ≈56 and ≈13 kDa and the β-subunit at ≈43 kDa, the purified SM deacylase was detectable only by the antibody to the β-subunit at ≈43 kDa. Breaking the S-S bond of recombinant human aCDase with dithiothreitol elicited the activity of SM deacylase with an apparent size of ≈40 kDa upon gel chromatography in contrast to aCDase activity with an apparent size of ≈50 kDa in untreated recombinant human aCDase. These results provide new insights into the essential role of SM deacylase as the β-subunit aCDase that causes the ceramide deficiency in AD skin.

## 1. Skin Characteristics and Barrier/Water Reservoir Functions in the Stratum Corneum of Patients with Atopic Dermatitis

Atopic dermatitis (AD) is a recurrent dermatitis with a high susceptibility to irritants and allergens, which is characterized clinically by severe dry skin even in nonlesional skin [1,2]. The nonlesional AD dry skin is thought to be a prerequisite factor for the easily provoked itching that results in accelerating the cutaneous permeability of many foreign substances due to the subsequent scratching, as has been called ‘scratch dermatitis’. The AD dry skin is also distinctly accompanied by both the cutaneous barrier disruption and an impaired water-holding capacity in the stratum corneum (SC) [3,4,5,6,7,8,9,10], features that contrast with xerosis [11] and contact dermatitis where there is no barrier abnormality in nonlesional skin, providing clinical insights into the pathogenesis of AD. The close relationship between the AD skin symptoms and SC functions has been well established in several clinical studies with AD patients including our group [3,4,5,6,7,8,9,10,12]. Thus, in one of our clinical studies with AD patients [4], comparison of SC functions with dry skin scores on the nonlesional AD skin demonstrated that while the capacitance values were highly correlated with dryness scores (*r* = −0.752, *p* < 0.0001, *n* = 106) and with scaling scores (*r* = −0.697, *p* < 0.0001, *n* = 106), the transepidermal water loss (TEWL) was also paralleled by dryness (*r* = 0.788, *p* < 0.0001, *n* = 106) with a higher correlation coefficient compared with capacitance values and scaling scores (*r* = 0.697, *p* < 0.0001, *n* = 106).

## 2. Abnormality in Percutaneous Permeability Barrier Function

Although TEWL is frequently used to measure the barrier function of skin, it is not necessarily a precise reflection of percutaneous permeability barrier, and thus we determined whether the chemical penetration rate is really increased or not in the nonlesional skin of AD patients compared with healthy control (HC) skin [3]. To detect in vivo cutaneous permeability, we used photoacoustic spectrometry (PAS) by which chemical concentrations present in SC layers up to 15 μm thick can be measured based on the intensity of photoacoustic signals derived from the chopped expansion of air due to chemical heat released during the relaxation process from chemical molecules excited by the chopper light. As penetrators, we utilized rhodamine B stearate (Red 215) and tartrazine (Yellow 4) as lipophilic and hydrophilic dyes, respectively, and determined the in vivo penetration rate of those lipophilic and hydrophilic dyes through the SC by photoacoustic signals that reflect chemical concentrations within the 15 μm thick SC layer. It turned out that both dyes penetrated faster in the nonlesional skin of AD patients compared with HC skin during the topical application period of 2 h, indicating that that there is a disruption in the in vivo cutaneous permeability barrier function against both the lipophilic and hydrophilic chemicals. To reduce the long time (such as 2 h) required to measure the disappearance rate of chemicals through the SC layers, we used patch chambers applied on the skin for 2 or 5 min and measured the concentrations of chemicals that penetrated into the SC layers, which are comparable with the penetration rates of chemicals. PAS analysis after application of the two dyes for 2 or 5 min under the closed patch conditions revealed that there was a significant increase (*p* < 0.001, *n* = 103 for AD and *n* = 10 for HC) in the photoacoustic signals equivalent to dye concentrations through the SC for both the lipophilic and hydrophilic dyes in the nonlesional AD skin compared with HC skin [3]. This result indicates that the nonlesional skin of AD patients has accelerated penetration rates for both types of dyes compared with HC skin. In this connection, recent systematic search for studies evaluating skin absorption of various penetrants indicated that patients with AD have almost twofold increased skin absorption compared with healthy controls [13].

## 3. Is Barrier Disruption a Cause or a Result of Dermatitis?

To determine if the barrier disruption is a cause or a result of dermatitis in AD nonlesional skin, we next compared the PAS signals of lipophilic and hydrophilic dyes with the severity of AD [2]. The difference between the PAS signals of the HC group (*n* = 10) and the mild AD group (*n* = 6) for the lipophilic dye (rhodamine B stearate) was significant (*p* < 0.05), which contrasts with an insignificant difference with the hydrophilic dye (tartrazine). On the other hand, the intensities of the PAS signals obtained with the hydrophilic dye (tartrazine) differed significantly among the three groups of AD patients in relation to the severity. The sum of these findings indicates that the intensity of the impaired percutaneous permeability barrier in the nonlesional AD skin is closely paralleled by the clinical severity of AD patients, which strongly suggests that the disruption of percutaneous permeability barrier homeostasis is generated as a result of the induced dermatitis and subsequently triggers a predisposition to evoke the recurrence of dermatitis.

Further, to elucidate whether the disrupted barrier function in nonlesional AD skin is associated with preinflammatory or postinflammatory events, which are relevant to the severity of AD, we evaluated the barrier function as measured by TEWL and the water content (conductance) as measured by an impedance meter of nonlesional AD skin and compared those SC functions with the severity of AD [4]. The TEWL was significantly increased in proportion to the severity of AD with a markedly high correlation coefficient (*r* = 0.834, *p* <0.0001, *n* = 106), while the capacitance decreased in proportion to the severity of AD with a relatively lower correlation coefficient (*r* = −0.720, *p* < 0.0001, *n* = 106). Taken together, the sum of our findings indicates that the barrier disruption and water deficiency in nonlesional AD skin are well suited to reflect the severity of AD and that nonlesional AD skin had already been inflamed and has never been purely normal skin since birth. Comparison between the TEWL and capacitance values in association with the AD severity indicated that those two parameters are well distributed depending on the severity of AD and that the elevated TEWL more adequately reflects the difference between HC skin and mild AD skin compared with the diminished capacitance (Figure 1) [4]. Therefore, our results indicate that the barrier disruption as well as the water deficiency in nonlesional AD skin reflect the disease severity of AD, which again suggests that the barrier abnormality as well as the water deficiency are elicited as a result of the evoked dermatitis in the past and subsequently trigger the recurrence of dermatitis. This combination also provides a useful insight into understanding the diagnosis and clinical improvement during therapy.

## 4. A Ceramide Deficiency Is Responsible for the Disrupted SC Function

Many studies have shown that the barrier disrupted dry skin of AD patients is mainly attributable to significantly decreased levels of total ceramides and altered ceramide profile in the SC [14,15,16,17,18,19,20,21,22,23]. The essential role of ceramides in expressing and maintaining these SC functions is strengthened by the findings of our group and other groups that ceramides can function as a water reservoir by holding bound water molecules that can never freeze and evaporate [24,25] and as a permeability barrier due to the formation of multilayered lamellar structures [26,27,28,29,30]. Further, the integrity of lipid lamellae in the SC of AD skin with the water deficiency and disrupted barrier homeostasis is distinctly lacking as a result of alterations in the ceramide profile, including the total ceramide level, its composite species and its alkyl chain properties [18,31].

## 5. Significance of Ceramides in SC Functions

The significant role of ceramides in SC water-holding and barrier functions is supported by direct clinical evidence that the disrupted barrier function and water deficiency that occurs in both the nonlesional and the lesional skin of AD patients [32] and in essential fatty acid-deficient mice or is elicited by surfactant or solvent treatment of HC skin can be distinctly restored to a healthy state by the topical application of natural ceramides or pseudo-ceramide (pCer) [33,34,35]. To provide deeper insights into the impaired barrier/water mechanisms that are associated with either ceramide or filaggrin-derived water soluble materials (mainly amino acids), a synthetic pCer that can function as natural ceramides and can be mass produced was designed and screened. This selection was based on its in vitro ability to form multilamellar structures and to retain bound-water molecules as well as its in vivo water-holding capacity and ameliorating effects on acetone-ether or surfactant-induced roughened skin [36,37,38]. We used pCer to elucidate if compensating ceramides or water-soluble materials could restore the perturbed barrier/water reservoir function in clinically normal, nonlesional AD skin. Thus, we conducted a clinical comparison study on nonlesional AD skin treated for 4 weeks with a 8% synthetic pCER cream or with a 0.3% mucopolysaccharide (HIRU) cream (used as a water-soluble material) [32], the latter of which is medical moisturizing cream approved for treating AD in Japan. Comparison of those two creams for clinical improvement revealed that whereas the HIRU cream elicited a slight improvement in 80% of subjects (*n* = 24) and a moderate improvement in 20% of subjects, the pCER cream resulted in a marked improvement in 50% of subjects, a moderate improvement in 36% of subjects and a slight improvement in 15% of subjects, which is significantly more efficient compared with the HIRU cream [32]. The pCER cream elicited a significant reduction in TEWL values (*p* < 0.0001) with significant and sharp decreases appearing at weeks 2 and 4. In contrast, the HIRU cream exhibited a significant but lesser reduction in TEWL values at weeks 2 and 4 compared with the pCER cream [32]. A comparison of the reduced TEWL values revealed that treatment with the pCER cream elicited a significantly greater reduction in TEWL (*p* < 0.0001) at week 4 compared with the HIRU cream. The pCER cream also induced a significant increase of 175% in capacitance values at weeks 2 and 4. In contrast, the HIRU cream elicited a significant but lesser increase of 140% in capacitance values only at week 2 compared with the pCER cream. A comparison of the increased capacitance values demonstrated that at weeks 2 and 4, treatment with the pCER cream elicited a significantly greater increase in capacitance values (*p* < 0.001) compared with the HIRU cream. As shown in Figure 1, when TEWL and capacitance values in association with disease severity were compared, rectangular areas representing each area consisting of means ± SD in TEWL and capacitance values were well distributed in association with severe, moderate and mild AD patients and HCs [11]. As shown in Figure 2, comparison between TEWL and capacitance values at 4 weeks of treatment in association with disease severity revealed that whereas those two parameters of pCER cream-treated skin were generally distributed into the rectangular areas corresponding to HC skin, those values for the HIRU cream-treated skin remain within rectangular areas corresponding to the mild or moderate AD skin [32]. Thus, this clinical study indicated a superior clinical efficacy of pCer and more importantly that the ceramide deficiency in the SC of AD skin plays a central role in the pathogenesis of AD although it remains unclear how much the altered ceramide profile in the decreased levels of total ceramides contributes to the disrupted SC function in AD skin.

## 6. Significance of the Ceramide Profile in SC Functions

As for the mechanistic contribution of altered ceramide profiles to the downregulation of SC water-holding and barrier functions, it remained to be clarified whether the changes in ceramide profile observed in AD skin [18,39,40] were atopic diathesis specific or were a mere facet that reflects inflammatory dermatitis such as chronic eczema or contact dermatitis because these related studies have no comparison with general non-AD as a disease control. Thus, Koyano et al. [41] reported that ceramide profiles in lesional but not in nonlesional SC of psoriasis patients resembles those of AD patients. Kim et al. [23] also recently demonstrated that the SC of nonlesional skin in patients with allergic contact dermatitis has abnormalities in barrier function and ceramide profiles that occur in a pattern similar to those in nonlesional AD skin. Thus, it seems reasonable to assume that the disrupted keratinization process that occurs subsequently due to cutaneous inflammation is mainly attributable to the ceramide profile changes, thereby exacerbating the pathological changes. This fact was also corroborated by our recent study which demonstrated that repetitive topical applications of pCer to AD skin significantly improved inflammation and atopic dry skin as well as the SC barrier/water reservoir function, accompanied by switching the ceramide profile to a healthy skin phenotype (Figure 3) [12]. Interestingly, these clinical and functional improvements in the SC can be achieved without any restoration of the decreased levels of total endogenous ceramides to a healthy state but with applied and compensated pCer accumulating at a similar level to existing endogenous ceramides in the SC. The remaining levels (µg/ng SC protein) of pCer applied to the SC of AD skin were significantly correlated (*n* = 39, *r* = 0.447, *p* = 0.005) with the increased water content measured by conductance whereas none of the ceramide species at the level of µg/ng SC protein were associated with the increased water content (Table 1). This suggests that total ceramide levels, including penetrated pCer in the SC, are more essential to restoring and maintaining the barrier and water reservoir functions than are the differential ceramide profiles and play an essential role in improving clinical skin symptoms. Thus, almost all topical application studies [5,6,7,8,9,10,12,32] of pCer on nonlesional AD skin demonstrated that compensating ceramide levels in AD nonlesional SC is significantly effective in improving atopic dry skin, accompanied by the amelioration of disrupted barrier and water holding functions as well as by switching the endogenous ceramide profile to a healthy skin phenotype [12]. These findings strongly suggest that the ceramide deficiency rather than altered ceramide profiles in the SC of AD skin plays an essential role in the pathogenesis of AD.

## 7. Is the Barrier Disruption and Its Associated Ceramide Deficiency Inherent or Not?

It is important to understand whether the impairment of skin barrier function and its associated ceramide deficiency in AD skin is inherent or not. In this respect, a prospective study of newborns revealed that the impairment of skin barrier function is not inherent with AD patients [42]. In adult AD skin, there is a ceramide deficiency even in the nonlesional SC, which is highly associated with the abnormal barrier function, and predisposes the skin to inflammatory processes evoked by irritants and allergens. Although it is possible that adult nonlesional AD skin may be postinflammation, it remains unclear whether the barrier disruption and ceramide deficiency in nonlesional AD skin results from postinflammation or from an inherited ceramide synthesis abnormality. To test this possibility, we studied infants with AD to examine SC functions and to compare SC ceramide levels and β-glucocerebrosidase (BGCase) activity in their skin [43]. TEWL in the forearm skin of infants with AD (*n* = 24~26) was significantly upregulated in the lesional skin but not in the nonlesional skin compared with HC skin (*n* = 18~20). When the relationship between TEWL and SCORAD index in the lesional and nonlesional skin of AD infants was assessed, there was a significant correlation (*n* = 17, *r* = 0.62, *p* = 0.013) between the two parameters in the lesional skin but not (*n* = 17, *r* = −0.08, *p* = 0.761) in the nonlesional skin, which suggests that the barrier disruption is not inherent but results from a subsequently elicited dermatitis. In contrast, the water content, as assessed by capacitance, was slightly but not significantly decreased only in the lesional skin. It is of great interest to note that ceramide levels, expressed as Mg ceramide/mg SC protein, were significantly decreased in the SC of lesional AD skin (*n* = 22) but not in the nonlesional AD skin (*n* = 28), which is consistent with the increased TEWL only in the lesional AD skin. When BGCase activity in the SC was evaluated, there was no significant difference between HC skin (*n* = 18~20) and lesional and nonlesional AD skin. In the non-lesional buttock skin of infants with AD (*n* = 26), there was no significant increase or decrease in TEWL values and water content, respectively, compared with HC skin (*n* = 18). The ceramide levels, expressed as Mg ceramide/mg SC protein, revealed that there is no significant decrease in SC ceramide in the nonlesional AD buttock skin which is consistent with the lack of an increase in TEWL values in the nonlesional AD buttock skin. BGCase activity in the SC of the non-lesional buttock skin demonstrated that there is no significant difference between the skin of HC and AD infants. In summary, as shown in Figure 4, in infant nonlesional AD skin, TEWL values and water content were not altered compared with HC infant skin. A comparison of ceramide levels demonstrates that SC ceramides are significantly reduced only in the lesional AD infant skin but not in the nonlesional AD infant skin compared with HC infant skin. On the other hand, there was no significant difference in BGCase activity in the SC of infant AD and infant HC skin as well as between adult AD and adult HC skin. These findings suggest that the barrier disruption due to the ceramide deficiency is not inherent and is essentially dependent on postinflammatory events in infants with AD. Although prevalent and rare loss-of-function mutations have been identified as the cause of the genodermatosis ichthyosis vulgaris and were additionally reported to be an important predisposing factor for the development of AD [44,45,46,47], our results on AD infant skin reflect the atopic diathesis as a genotypic predisposition but are not consistent with loss-of-function mutations in the gene encoding filaggrin in AD which may represent only one facet of the atopic abnormalities.

## 8. Clinical Evidence for the Impaired Homeostasis of the Ceramide-Generating Process

Thus, the evidence for the involvement of inflammation in the predisposition to a ceramide deficiency which results in impaired barrier and water reservoir functions prompted us to elucidate the effects of evoked inflammation on barrier/water function and ceramide biosynthesis in the SC of AD patients. We used tape-stripping to induce cutaneous acute inflammation and compared changes in the levels of barrier disruption and water content as well as of ceramides and sphingolipid enzyme activities between AD and HC skin during the barrier recovery process [21]. A similar approach with tape-stripping has been challenged in Nieman–Pick patients who have acid sphingomyelinase (aSMase) deficiency, which showed that the delayed barrier recovery is mainly ascribed to the inherently downregulated levels of aSMase activity due to mutation of the aSMase gene [48]. Based on this evidence, if the alteration of some ceramide-metabolic enzymes or related enzymes are involved as causative factors in the continued barrier disruption of AD skin, this challenge would provide a deep insight into phenotypic changes of barrier function and ceramide content in the SC of AD patients in response to cutaneous inflammation as well as into unknown factors that may be provoked following acute inflammation. Our results showed that basal levels of capacitance values prior to the tape-stripping are significantly lower in AD skin than in HC skin. During the tape-stripping process, the water content (capacitance value) increased with increased numbers of tape-strippings to a lower extent in AD skin than in HC skin. During the recovery process, at 4 days post-stripping, capacitance values in AD skin returned to almost the same values as before the tape-stripping. In contrast, capacitance values in HC skin returned to less than the pre-tape-stripping level. The skin capacitance values then returned to similar levels between AD and HC skin at 4 days post-stripping. On the other hand, basal levels of TEWL values prior to tape-stripping were significantly higher in AD skin than in HC skin. During the tape-stripping process, TEWL values increased with increasing numbers of tape-strippings to a similar extent in AD skin and in HC skin. During the recovery process, at 4 days post-stripping, TEWL values both in AD skin and in HC skin returned to higher levels than before the tape-stripping, the level of which was significantly higher in AD skin than in HC skin. The recovery of TEWL values after tape-stripping occurred at a significantly slower rate at 1 day post-tape-stripping in AD skin than in HC skin, but proceeded at a similar rate at 2, 3 and 4 days post-tape-stripping (Figure 5). Since a similar recovery rate was observed in Niemann–Pick disease caused by aSMase deficiency with a delay of barrier recovery only at 1 day post-tape-stripping [48], these tape-stripping studies on AD skin strongly suggest that the impaired homeostasis of a ceramide-generating process occurs in AD skin.

In a parallel study, thin layer chromatographic analysis demonstrated that basal levels of ceramides in the SC of AD skin significantly differ from HC skin with significantly lower levels in AD skin than in HC skin (Figure 6). Comparison of ceramide levels in the SC between before and after tape-stripping revealed that whereas ceramide levels in HC skin are significantly upregulated at 4 days post-tape-stripping, the ceramide levels in AD skin remain substantially unchanged at 4 days post-tape-stripping (Figure 6). Analysis of ceramide species indicated that basal levels of ceramides [NP], [AS], [NH], [AP], and [AH] are significantly lower in AD skin than in HC skin [21]. Comparison of ceramide species before and after tape-stripping revealed that levels of ceramides [EOS], [NP], [AP], and [AH] are significantly increased at 4 days post-tape-stripping in HC skin whereas levels of all ceramide species are unchanged in AD skin [21]. When BGCase activity was assessed, basal levels of the enzyme activity in the SC occurred at a similar level between AD skin and HC skin [21]. Comparison of the BGCase activity in the SC before and after tape-stripping demonstrated that whereas BGCase activities at 4 days post-tape-stripping are significantly upregulated in HC skin compared with before tape-stripping, those activities in AD skin remain substantially unchanged [21]. While BGCase activity showed more than a 144% increase in HC skin at 4 days post-tape-stripping, BGCase activity was not upregulated (<a 97% increase) in AD skin with a significant difference in % increase between the two. As there is no significant difference in the basal levels of BGCase activity between HC skin and nonlesional and lesional AD skin, this strongly suggests that the failure to stimulate BGCase activity is mainly attributable to the tape-stripping triggering unknown abnormal biological events. Thus, this may occur via possible abnormal pathways through which levels of ceramide precursors, sphingomyelin (SM) as well as glucosylceramide (GCer), are also not converted at a sufficient and healthy level by the corresponding hydrolytic enzymes acid sphingomyelinase (aSMase) and BGCase, respectively, to their hydrolytic end-product ceramide.

Ceramide levels in the SC are modulated by the balance of three enzymes involved in sphingolipid hydrolysis, BGCase, aSMase and acid ceramidase (aCDase), secreted in lamellar granules (LGs) in the interface between the stratum granulosum and the SC. Therefore, as for the biological mechanisms involved in the ceramide deficiency in AD skin, it is important to elucidate the activities of those three sphingolipid hydrolysis enzymes in the SC or the epidermis of AD skin compared with HC skin. Two of those three sphingolipid hydrolysis enzymes, aSMase and BGCase, were not attenuated at the enzymatic activity level [17,49,50] or at the protein level [51,52] in the nonlesional epidermis or the nonlesional SC of AD skin. Although one study did report a decreased activity of aSMase [53], the deficiency of aSMase activity only cannot reasonably account for the ceramide deficiency in AD skin because of the diminished levels of all ceramide species including acylceramide, a ceramide species that is not synthesized by aSMase. Further, other sphingolipid metabolic enzymes that function upstream of the hydrolysis of GCer and SM in the epidermis, such as serine-palmitoyl transferase (SPT), stearoyl CoA desaturase (SCD), ceramide synthases (CERS) 1–5, GCer synthase (GCERS), alkyl chain elongation enzymes and SM synthase (SMS), have never been reported to be implicated in the ceramide deficiency in uninflamed nonlesional AD skin [52]. Since a barrier recovery pattern with a significant delay only at 1 day after tape-stripping in AD skin compared to HC skin was almost identical to that as seen in aSMase-deficient Niemann–Pick disease, we thought it likely that the impaired homeostasis of a ceramide-generating process other than the known sphingolipid metabolic enzymes may be associated with the continued abnormality of barrier and water reservoir functions due to the ceramide deficiency in nonlesional AD skin. Thus, it is intriguing to know what biological factors would trigger the epidermis to downregulate the synthesis of SC ceramides in AD skin.

## 9. Discovery of Sphingomyelin Deacylase

While there is no abnormality in the activity or the expression of ceramide-generating enzymes, including aSMase and BGCase, in AD skin [17,21,49,50], we analyzed SM hydrolysis in detail using radio-thin layer chromatography (RTLC) with [choline-methyl-^14^C]SM. RTLC analysis for the first time revealed that radiolabeled SPC was enzymatically released following incubation with the SC lysate from AD skin but not from contact dermatitis skin or HC skin (Figure 7) [54]. There are three possible ways by which radiolabeled materials could be released enzymatically from [choline-methyl-^14^C]SM, as shown in Figure 8: (1) the formation of [^14^C]PC by SMase; (2) the formation of [^14^C]sphingosylphosphorylcholine (SPC) by a deacylase-like enzyme; (3) the formation of [^14^C]choline by a phospholipase D-like enzyme.

The observed release of radiolabeled SPC from [choline-methyl-^14^C]SM strongly suggested that the presence of a deacylase-like enzyme (SM deacylase) in the SC of AD skin. Therefore, as depicted in Figure 9, we thought it likely that the high expression level of SM deacylase competes with the ceramide-generating enzyme aSMase for the same substrate SM, which results in the decreased levels of ceramides seen in the SC of AD patients. Another study using RTLC analysis (Figure 10) demonstrated that similarly radiolabeled SPC is released from [choline-methyl-^14^C]SM following incubation with a SM deacylase-rich fraction (measured using [fatty acid-^14^C]SM as a substrate), which was partially purified from lysates of the SC of AD skin by analytical isoelectric focusing (IEF) chromatography (Figure 11) [55]. As the SM deacylase-rich fraction isolated by preparative IEF chromatography did not have any activities of aCDase, aSMase or BGCase (Figure 11) [55], these findings strongly indicated that SM deacylase enzymes exist in the SC of AD skin that are distinct from aCDase or other enzymes.

In a further study [55] to characterize the properties of SM deacylase, its activity showed a pH dependency with a peak at pH 5.0. As shown in Figure 10, analytical IEF chromatography demonstrated that the pI values of SM deacylase, aSMase, BGCase, and aCDase are 4.2, 7.0, 7.4, and 5.7, respectively. Enzymatic assays with N-[palmitoyl-1-^14^C]SM as a substrate following gel chromatography revealed that the activity of gel-fractionated SM deacylase, detectable by the release of ^14^C-free fatty acid, migrates with a molecular weight of ≈40,000, while aSMase, detectable by the release of ^14^C-ceramide, migrates at a molecular weight of ≈100,000 (Figure 12) [55]. Figure 12 shows preparative SDS-PAGE of epidermal extracts from AD patients. The gel was sliced into 20 pieces, homogenized and subjected to measurements for SM hydrolysis. In the assay for SM hydrolysis with N-[palmitoyl-1-^14^C]SM (Figure 13A) or [choline-methyl-^14^C]SM (Figure 13B) there were two separated peaks of radioactivity distributed in the lipophilic or aqueous phases, respectively, each corresponding in electrophoretic mobility to the molecular weights estimated for SM deacylase and aSMase. Furthermore, RTLC analysis of reaction products generated by fraction peak II demonstrated that the enzyme possesses activity that can release radiolabeled SPC (Figure 12B).

Taken together, it is likely that the enzymatic characteristics of SM deacylase account for the deficiencies in some species of ceramides seen in the SC of AD skin [12], which provides an etiological basis for the dry and barrier-disrupted skin of AD patients.

## 10. SM Deacylase Activity in AD Skin

For the characterization of possible mechanisms involved in the ceramide deficiency of AD skin, a new quantitative assay for SM was established. As shown in Figure 14, our quantitative measurements [50] clearly demonstrated that SM deacylase activity is enhanced more than 5-fold in lesional SC and more than 3-fold in nonlesional SC of AD skin, compared with the SC of HC skin. In contrast, the SC from patients with contact dermatitis showed no increase in SM deacylase activity compared with HCs, which suggests that changes in SM deacylase activity are unlikely to be involved in the general etiology of cutaneous inflammation. Our earlier study on the epidermal localization of aCDase and BGCase, the latter being a hydrolytic enzyme localized in intercellular spaces between the SC and the granular layer, suggested that the activities of ceramide metabolism-related enzymes within the SC approximately represent the epidermal activities of the same enzymes [56,57,58]. Consistent with that relationship, a similar higher level of SM deacylase activity was detected in the epidermis from AD skin (Figure 15), whereas there was no significant difference in levels of aSMase between AD skin and HC skin [50], which suggests that epidermal cells from AD patients show abnormal production of the hitherto undiscovered epidermal enzyme termed SM deacylase.

## 11. GCer Deacylase Activity in AD Skin

The specific expression of SM deacylase in AD skin provides a strong basis to explain the ceramide deficiency. However, since acylceramides are not generated through SM metabolism but are markedly decreased in the SC of AD skin, that explanation of altered SM metabolism leading to the ceramide deficiency is not the complete story. Acylceramide is a unique species of the ceramide family that has been shown to be an essential component involved in the barrier homeostasis since it is involved in forming the multilamellar membranous architecture in intercellular spaces between the SC layers [59]. The importance of acylceramide in maintaining the barrier function was also supported by our previous studies [60]. Thus, topical application of pCer containing ester-linked linoleic acid on the skin of essential fatty acid-deficient rats was found to restore the disturbed barrier function revealed by increased TEWL and epidermal hyperplasia. In AD skin, the acylceramide deficiency was found to exist to the highest degree among several ceramide species, being predominantly attributable to the constitutive barrier disruption seen even in nonlesional AD skin [14]. Topical application of pCer-containing ester-linked linoleic acid on the skin of AD patients completely restored the skin barrier disruption and recovered TEWL values to HC levels [61]. Therefore, it is of considerable importance to determine the biochemical mechanism(s) involved in the downregulation of acylceramide production in order to elucidate the pathogenesis of AD and its predisposition toward recurrent dermatitis.

Our related study [62] using specific inhibitors of glucosyltransferase and BGCase, suggested that acylceramides are eventually generated through the deglycosylation of acylGCer by BGCase. In our biochemical study using [^14^C-palmitic acid]GCer, we found that the pI 4.2 IEF fraction, which contains partially purified SM deacylase, with no contaminating aCDase, BGCase or aSMase, can hydrolyze [palmitic acid-^14^C]GCer, but not N-[palmitoyl-1-^14^C]Cer at its acyl site to yield ^14^C-free fatty acid (Figure 16 and Figure 17) [55].

This finding strongly suggests that, as shown in Figure 18, SM deacylase can compete with BGCase for the same substrates, GCer or acyl-GCer. This would yield glucosylsphingosine (GSP) rather than ceramide, which would in turn lead to a deficiency of ceramides, including acylceramides, since they are generated through the deglucosylation of acyl-GCer by BGCase.

Using [palmitic acid-^14^C]GCer as a substrate, we measured the activity of GCer deacylase in the SC and in the epidermis of AD skin by the potential to directly hydrolyze GCer at the *N*-acyl site to release ^14^C-labeled free fatty acid and compared that with levels found in HC skin As shown in Figure 19, comparison of the activity of GCer deacylase revealed that there are significant increases in its activity in the lesional (5.4-fold increase) and in the nonlesional (2.5-fold increase) SC from AD skin or in the lesional (3-fold increase) epidermis of AD skin compared with HC skin [17]. Collectively, our observations suggested that the acylceramide deficiency in AD is mainly attributable to the accentuated activity of GCer deacylase. Thus, the abnormally expressed GCer deacylase hydrolyzes (acyl) GCer at the N-acyl site to yield its lysoform, GSP, instead of the formation of (acyl)ceramides by BGCase. The enhanced activity of this novel enzyme is proportional to the diminished level of (acyl)ceramide because of the enzymatic competition toward the same substrate, GCer, between GCer deacylase and BGCase.

## 12. Accumulation of SPC as Evidence for Functional SM Deacylase

To clarify whether SM deacylase is functioning in situ in the epidermis of patients with AD to yield SPC instead of ceramide, it is important to determine whether its enzymatic reaction product, SPC, is released into the interface between the granular and the SC layers and accumulates in the SC.

Quantitative analysis of SPC in the SC of AD skin revealed that there was a significant increase in the content of ng SPC/mg SC in nonlesional and in lesional SC compared with age-matched HCs (Figure 20) [20]. In contrast, there was no increase in SPC content in the lesional SC of patients with chronic eczema, which suggests that the upregulation of SPC in AD skin does not result from ordinary inflammation, but is associated with the altered lipid metabolism characteristic of AD.

Comparison between the ceramide and SPC levels in the same individuals demonstrated that there is a weak inverse relationship between levels of ceramides and SPC that accumulate in the SC (Figure 21) [20]. In contrast, comparison with sphingosine (SS), which is a degradative product from ceramide produced by aCDase, demonstrated that there is no correlation (*n* = 32, *r* = −0.182, *p* = 0.319) between levels of SPC and SS in the overall group as well as in the individual groups [20]. In the groups used for correlation analysis there were significant decreases in the amounts of SS in the nonlesional and lesional SC from patients with AD compared to HC skin.

As depicted in Figure 22, which shows the relationships between enzyme defects and the accumulation of enzymic reaction products, a similar accumulation of SPC has been observed in Niemann–Pick disease [63]. Niemann–Pick disease is associated with defects in aSMase, which results in the lipidosis for SM [48], although no data has suggested that this accumulation of SPC is linked to the expression of an SM deacylase-like enzyme. Since SPC accumulation has been speculated to result from a defect of aSMase (because SPC is a substrate for aSMase) [48], it would be intriguing to determine whether there is a similar upregulated expression of SM deacylase in Niemann–Pick disease that could provide a mechanism for the production of SPC in that disease as it does for AD [50,55].

## 13. Accumulation of GSP as Evidence for Functional GCer Deacylase

To characterize the physiological and functional relevance of GCer deacylase to the acylceramide deficiency in the epidermis of patients with AD, it is important to determine whether its enzymatic reaction product, GSP, is released into the epidermis and accumulates in the SC. Quantitative analysis of GSP using the radiolabeling technique in the upper SC of patients with AD revealed that there is a significant increase in the content of ng GSP/mg SC in both the uninvolved and the involved SC of AD patients compared with age-matched HC skin (Figure 23) [17].

Comparison of the amounts of total ceramides and ceramide-1 with GSP in the same individuals demonstrated that there is a weak inverse relationship (*n* = 36, *r* = −0.416, *p* < 0.05 for total ceramides and *n* = 52, *r* = −0.5243, *p* < 0.001 for Cer[EOS]) between levels of ceramides and GSP that accumulate in the SC (Figure 24) [17]. In comparison with SPC, there was a significant positive correlation between levels of GSP and SPC (Figure 25)**.**

As depicted in Figure 22 to explain the relationship between enzyme defects and the accumulation of enzymatic reaction products, a similar accumulation of reaction products by corresponding *N*-deacylase enzymes has been found in Gaucher disease, in which there is an accumulation of GSP due to a defect of BGCase activity [64]. Thus, the accumulation of GSP has also been speculated to result from a defect of BGCase since GSP can serve as a substrate for that enzyme [65]. However, GSP has already been established as a key biomarker of Gaucher disease [66]. Similarly, the possible existence and expression of GCer deacylase in Gaucher disease would provide a reasonable mechanism for the upregulation of GSP. Another similar relevance of *N*-deacylase for the generation of psychosine has been reported in globoid cell leukodystrophy (GLD) or Krabbe’s disease [67]. The primary defect of GLD is a deficiency in galactosylceramidase activity, which leads to the accumulation of galactosylceramide and its metabolic intermediate galactosylsphingosine. This was speculated to be produced by the deacylation of galactosylceramide [67], although there is no evidence for the expression of galactosylceramide deacylase in GLD. Such altered lipid metabolisms associated with genetic defects, which lead to the accumulation of lipid substrates and deacylated metabolic intermediates, strongly suggest the principle that defects of metabolic enzymes might induce corresponding alternative pathways in which those substrates are converted to corresponding lysoforms by deacylation. Such a possible induction of an alternative pathway following the loss of metabolic enzymes has been reported in a Gaucher-like mouse model induced by a glucosylceramidase inhibitor which causes the accumulation of GSP in tissues [64]. A similar relationship between the inhibition of BGCase and the accumulation of GSP has been reported in fibroblasts using conduritol β-epoxide (CBE) [65]. Thus, when CBE was added to the culture medium, the intracellular β-glucosidase activity decreased, and both GCer and GSP accumulated in the cells. Based on this evidence, it has been suggested that the synthetic pathway for GSP is considered to not only be through the glucosylation of SS, but also through the deacylation of GCer [65].

In conclusion, as depicted in Figure 26, we elucidated the functional relevance of SM GCer deacylase to the ceramide deficiency that is an essential etiologic factor for the dry and barrier-disrupted skin of patients with AD. Interestingly, the enzymatic reaction products, SPC and GSP, which are essential surrogates to determine whether SM GCer deacylase is functioning in situ in the epidermis, is significantly increased in the nonlesional and lesional SC of patients with AD compared with HCs, and is reciprocally related to the decreased levels of ceramides in a similar group of patients with AD.

## 14. Altered Sphingolipid Metabolism May Contribute to Atopic Skin Phenotypes (Inflammation, Roughened Skin and Hyperpigmentation)

Regarding the biochemical connection and clinical loop among the ceramide deficiency, barrier disruption, altered sphingolipid metabolism and atopic skin phenotypes, it was of considerable interest to determine how the biologic functions of human epidermal cells such as keratinocytes and melanocytes are affected by SPC with respect to the excess formation of SPC in the epidermis of AD patients and their high susceptibility to inflammation. Therefore, we characterized the roles of SPC in human epidermis by elucidating its biologic effects on the expression of intercellular adhesion molecule-1 (ICAM-1) and a keratinization enzyme transglutaminase (TGase) by human keratinocytes as well as on the expression of melanogenic factors by human melanocytes in comparison with other sphingolipids. As depicted in Figure 27, it turned out that SPC metabolites due to the enzymatic action of SM deacylase, induce the expression of ICAM-1 [68] and activate TGase in human keratinocytes [69] and accelerate melanin synthesis in human melanocytes [70]. These findings strongly indicate that the phenotypes of AD skin, such as inflammation, roughened skin and hyperpigmentation are at least in part associated with altered sphingolipid metabolism.

## 15. Identification of SM Deacylase at the Gene and Protein Levels

The sum of the above findings indicates that the specific expression of SM deacylase activity in AD skin provides a reasonable hypothesis to explain why the level of ceramide in the SC continues to be significantly downregulated and is not upregulated even by frequently induced inflammation in the SC of nonlesional AD skin in concert with the lack of any substantial attenuation of the three major ceramide-related hydrolytic enzymes, the abnormalities of which markedly contrast with HC skin. Therefore, it was intriguing to purify the SM deacylase to homogeneity and to identify the detailed characteristics of this novel enzyme at the gene and protein levels.

## 16. Purification of SM Deacylase

To precisely measure SM deacylase activity, we developed a highly sensitive LC-MS/MS method which has the capacity to accurately measure the reaction product, SPC, over five orders of magnitude [71]. Since there was a distinct activity of SM deacylase in extracts after the centrifugation of homogenates of normal rat skin, we purified SM deacylase to homogeneity (a single spot by 2D-SDS-PAGE) (Figure 28) after a five-step purification procedure consisting of hydrophobic chromatography, IEF chromatography, ion-exchange chromatography, gel filtration chromatography, and immune-affinity chromatography. The purified SM deacylase had an apparent molecular mass of 43 kDa, an enrichment of >14,000-fold, and maximal pH and pI values of 5.0 and around 7.0, respectively [71].

## 17. Enzymatic Properties of Purified SM Deacylase

The purified SM deacylase followed normal Michaelis–Menten kinetics with V_max_ and K_m_ of 14.1 nmol/mg/h and 110.5 µM, respectively [71]. These enzymatic properties of pH dependency and molecular weight are consistent with those properties (pH = 4.7, MW = 40 kDa) obtained in our previous study of AD skin [55]. However, the pI value of the purified SM deacylase was different from our previous report using analytical IEF chromatography of a homogenate of the SC of AD skin, which detected pI values of SM deacylase, BGCase, aSMase, and aCDase of 4.2, 7.4, 7.0, and 5.7, respectively [55]. Thus, the pI value of SM deacylase was markedly shifted from 5.0 in the skin homogenate to 7.0 in the purified enzyme with an enhancement of activity by ≈200-fold after IEF chromatography. That result suggested a conjugation of SM deacylase enzyme with a natural inhibitory subunit with acidic pI molecules.

## 18. Identification of SM Deacylase at the Protein Level

Analysis by MALDI-TOF MS/MS revealed that the single protein spot with SM deacylase activity separated by 2D-SDS-PAGE was the β-subunit of aCDase, an enzyme consisting of α- and β-subunits linked by amino-bonds (Cys-143 /Met-142 in rat; /Ile-142 in human) and a single S-S bond(C31/C340) (Figure 29).

In Western blotting analysis using our novel β-subunit specific antibodies under reduced or nonreduced conditions, the SM deacylase purified from rat skin had a distinct band of ≈40 kDa (Lane 1) which was consistent with the band detected for recombinant human aCDase (Lane 2) under reduced conditions (ME+) (Figure 30). These results confirmed that the single protein spot with SM deacylase activity separated by 2D-SDS-PAGE is identical to the β-subunit of aCDase. This identification was also corroborated by a gel chromatographic analysis demonstrating that breaking the disulfide bond (C31/C340) of recombinant human aCDase with the reducing agent dithiothreitol (DTT) provokes the activity of SM deacylase with ≈40 kDa upon gel chromatography (Figure 31) [71].

aCDase is a lysosomal enzyme that is present especially in the epidermis including the SC [15,55] and catalyzes the hydrolysis of ceramides into fatty acids and SS. In the skin, SS production is associated with SSP-related signaling in keratinocytes [72] as well as in the ceramide-degrading process in the SC [15]. The aCDase protein had been purified to apparent homogeneity from urine [73] and placenta [74] and its full-length cDNA sequence was determined [75]. In human and rat aCDase, autoproteolytic cleavage has been documented to occur between Cys-143 and Ile-142 (Met-142 in rats) [76,77,78,79]. As for the natural activation mechanism of aCDase, a study using a crystal aCDase model hypothesized that the three dimensional configuration of the substrate binding channel in activated aCDase after autocleavage is specific for ceramide, as acyl-residue-containing sphingolipids with bulky head groups such as SM and GCer would be sterically hindered and unable to work as a substrate for aCDase [77]. Thus, it seems reasonable to assume that both the autocleavage and the breaking of the disulfide bond between the α- and β-subunits are essential requirements for the expression of SM deacylase. We thought it likely that in the epidermis of HC skin, the proenzyme aCDase undergoes autocleavage into α- and β-subunits in the intracellular lysosomal system without breaking the S-S bond between the α- and β-subunits to acquire aCDase activity. In contrast, in the epidermis of AD skin, it seems likely that the β-subunit is generated both by auto-cleavage of the covalent peptide bond between Ile-142 in the α-subunit and Cys-143 in the β-subunit and by breaking the S-S bond (C31/C340) between the α- and β-subunits of aCDase via unknown mechanisms, which leads to the induction of SM deacylase activity. Thus, it is probable that both cleavages result in removing the steric hindrance in the enzymatic active pocket against acyl-residue-containing sphingolipids with bulky head groups such as SM and GCer and leads to the expression of the activities of SM deacylase and GCer deacylase, which occur as enzymatic deacylation reactions in the same active pocket as aCDase.

These uncovered enzymatic alterations of aCDase in AD skin also provide a deep insight into understanding the pathogenesis of Gaucher disease, Fabry disease and Niemann–Pick disease where GSP, globotriaosylsphingosine and SPC, respectively, are formed due to deficiencies of BGCase (Gaucher disease), α-galactosidase (Fabry disease) and SMase (Niemann–Pick disease). Thus, the molecular basis for the formation of corresponding lysosphingolipids by deacylation reactions is attributable to the altered enzymatic properties of aCDase, which can trigger the deacylation of GCer, galactosylceramide and SM. In fact, one recent study [80] pointed to an active role of aCDase in the former two processes through deacylation of lysosomal glycosphingolipids although the contribution of aCDase was confirmed only by the inhibitory effect of an inhibitor of aCDase on the formation of GSP and globotriaosylsphingosine. It is possible that the inhibitor can also affect GCer/SM deacylase.

Based upon the above findings on SM deacylase, as shown schematically in Figure 32, we hypothesize two possible causative biological factors that underlie the expression of SM deacylase in AD skin as follows: (1) the formation of the S-S bond (C31/C340) between the α- and β-subunits of aCDase could be impaired in AD skin, probably due to a point mutation of the aCDase proenzyme although no such point mutations are currently known; (2) breaking the S-S bond (C31/C340) could occur more easily in AD skin than in HC skin via unknown mechanisms.

## 19. Conclusions

In conclusion, our finding that the pathogenic ceramide degrading enzyme SM deacylase, discovered as a causative factor that downregulates ceramide synthesis in the SC of AD skin, is identical to the β-subunit of aCDase provides an essential and deep insight into understanding the pathogenesis of AD. This should facilitate therapeutic approaches for developing specific inhibitors of SM deacylase that could be applied topically or orally to essentially abrogate the ceramide deficiency in AD skin, which would result in the essential cure of AD even in the chronic phase, which contrasts with recent therapeutic drugs that mainly target immunological aspects in the acute phase [81].

## Figures and Tables

**Figure 1 ijms-22-01613-f001:**
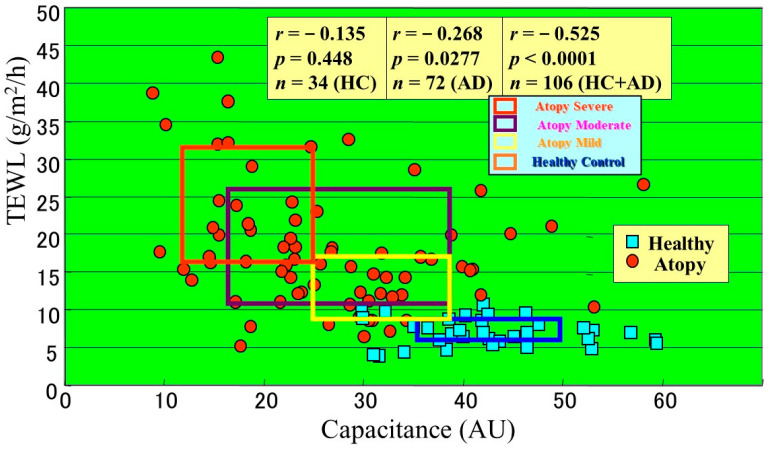
Relationship between TEWL and capacitance values in association with the severity of AD [4].

**Figure 2 ijms-22-01613-f002:**
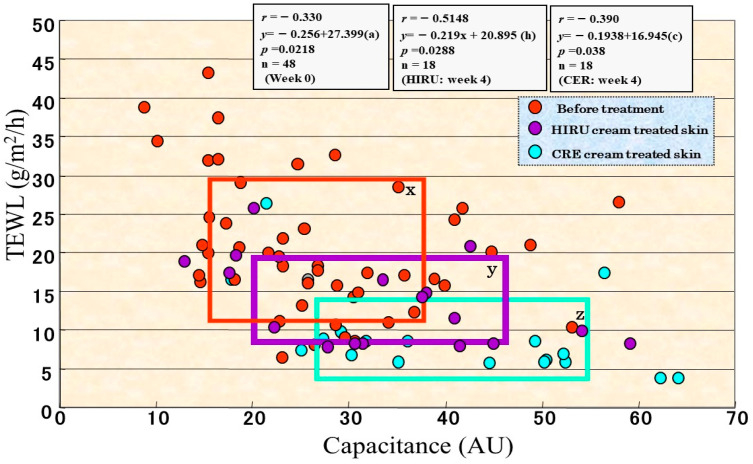
Comparison between TEWL and capacitance values during 4 weeks of treatment with pCER or HIRU creams in association with disease severity [32]. x: severe, y: mild, z: healthy control in disease severity.

**Figure 3 ijms-22-01613-f003:**
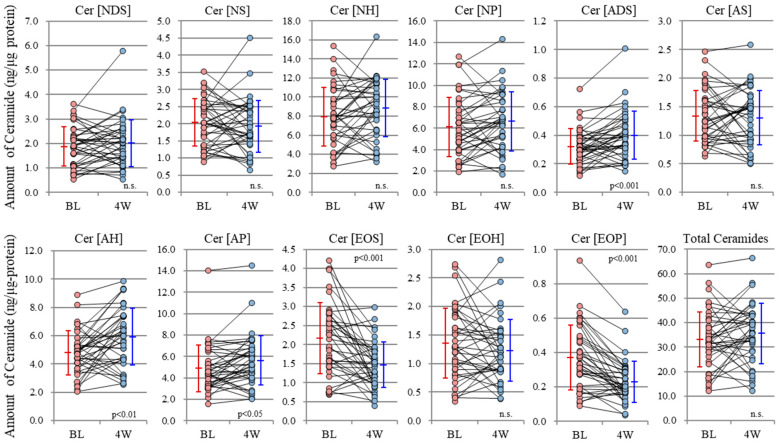
Changes in the class ratio (ng/μg protein) of endogenous ceramides in the SC of nonlesional AD skin after 4 weeks of using the pCer lotion (*N* = 39) [12].

**Figure 4 ijms-22-01613-f004:**
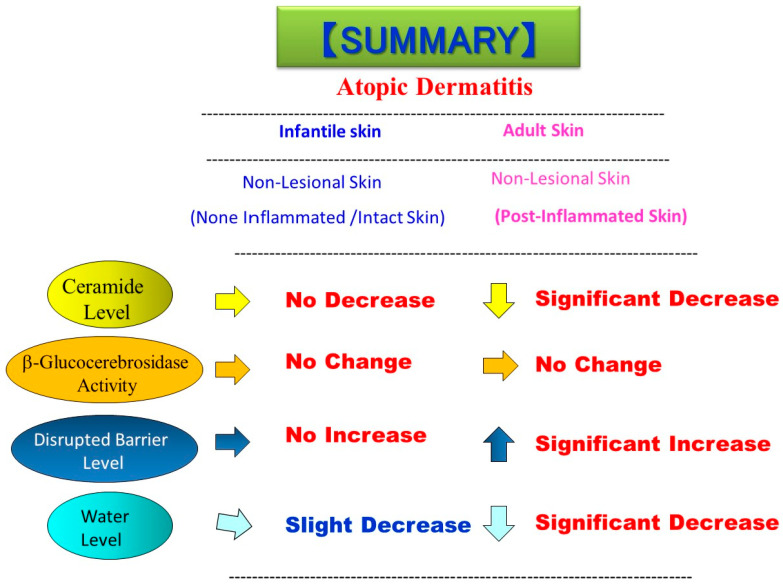
Comparison of the levels of ceramide, BGCase activity, barrier function, and water content between the nonlesional skin of infants and adults with AD [43].

**Figure 5 ijms-22-01613-f005:**
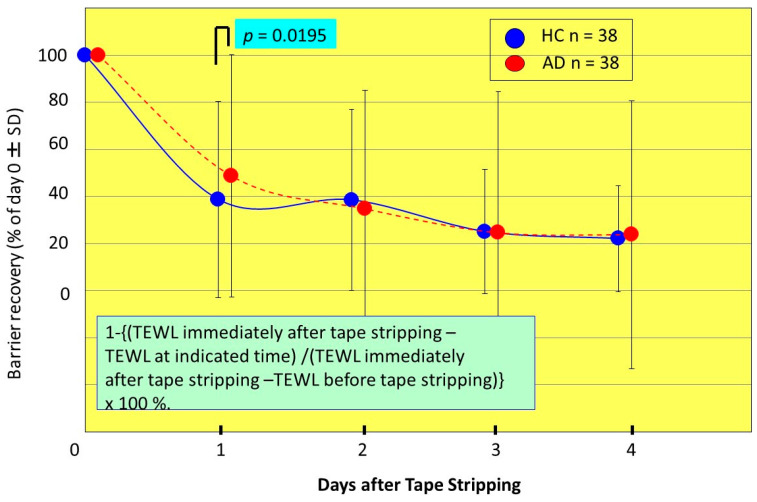
Percent recovery in TEWL values following tape-stripping [21].

**Figure 6 ijms-22-01613-f006:**
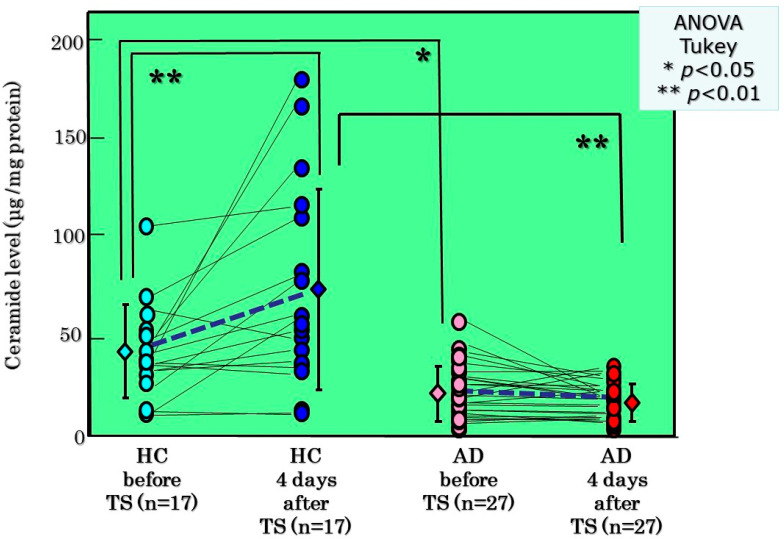
TLC analysis and changes in ceramide levels by tape-stripping [21].

**Figure 7 ijms-22-01613-f007:**
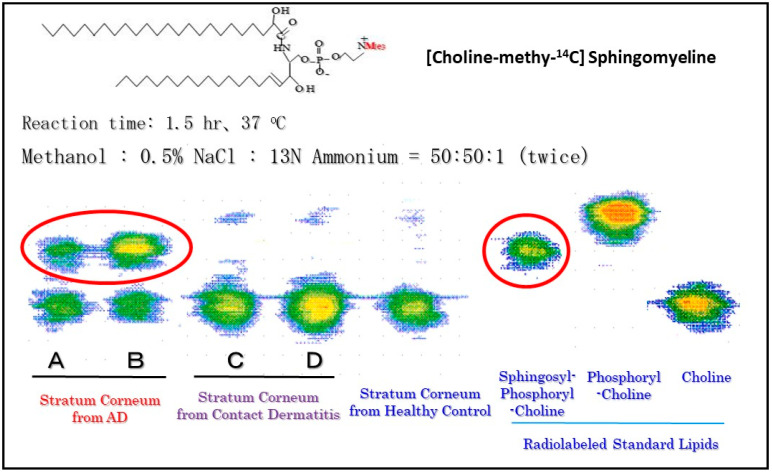
Radio-TLC analysis of reaction products following incubation of [choline-methyl −^14^C]SM as a substrate with samples from the SC [54].

**Figure 8 ijms-22-01613-f008:**
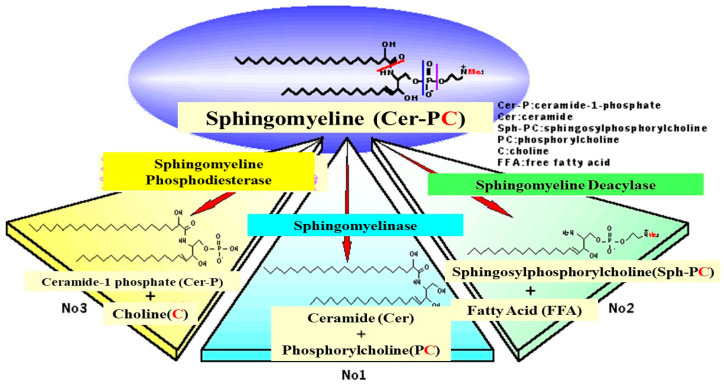
Three possible mechanisms for the hydrolysis of SM by natural enzymatic reactions [54].

**Figure 9 ijms-22-01613-f009:**
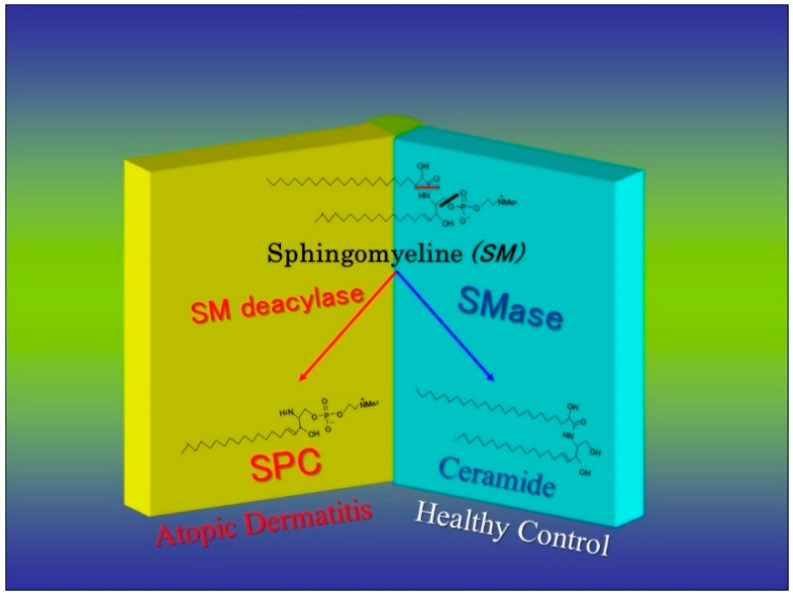
Enzymatic scheme of SM deacylase [50,54].

**Figure 10 ijms-22-01613-f010:**
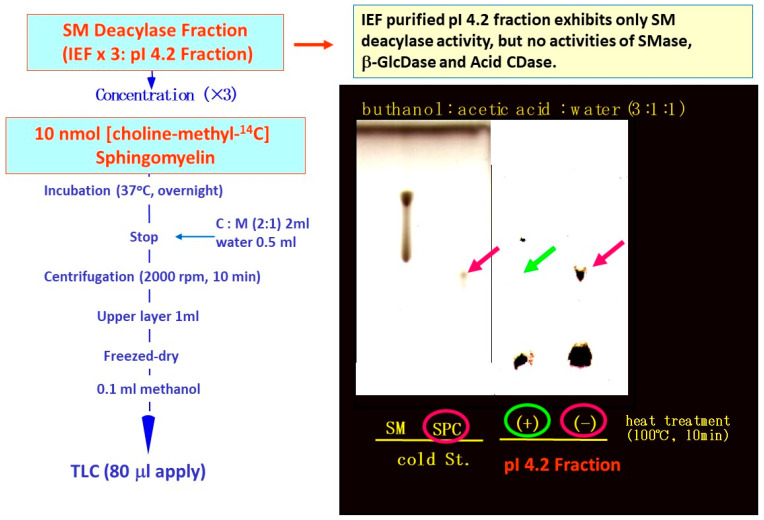
RTLC analysis using [choline-methyl-^14^C]SM as a substrate following incubation with SM deacylase-containing fractions that had been partially purified from extracts of AD SC by analytical IEF chromatography [55].

**Figure 11 ijms-22-01613-f011:**
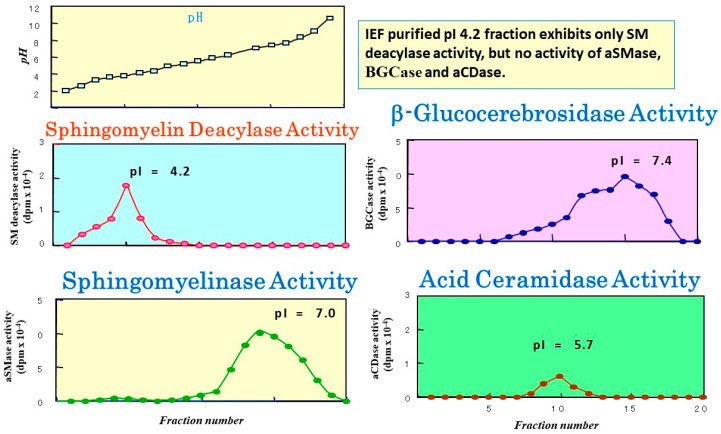
Analytical IEF chromatography using the atopic SC for SM deacylase, aSMase, BGCase, and aCDase and corresponding pI values [55].

**Figure 12 ijms-22-01613-f012:**
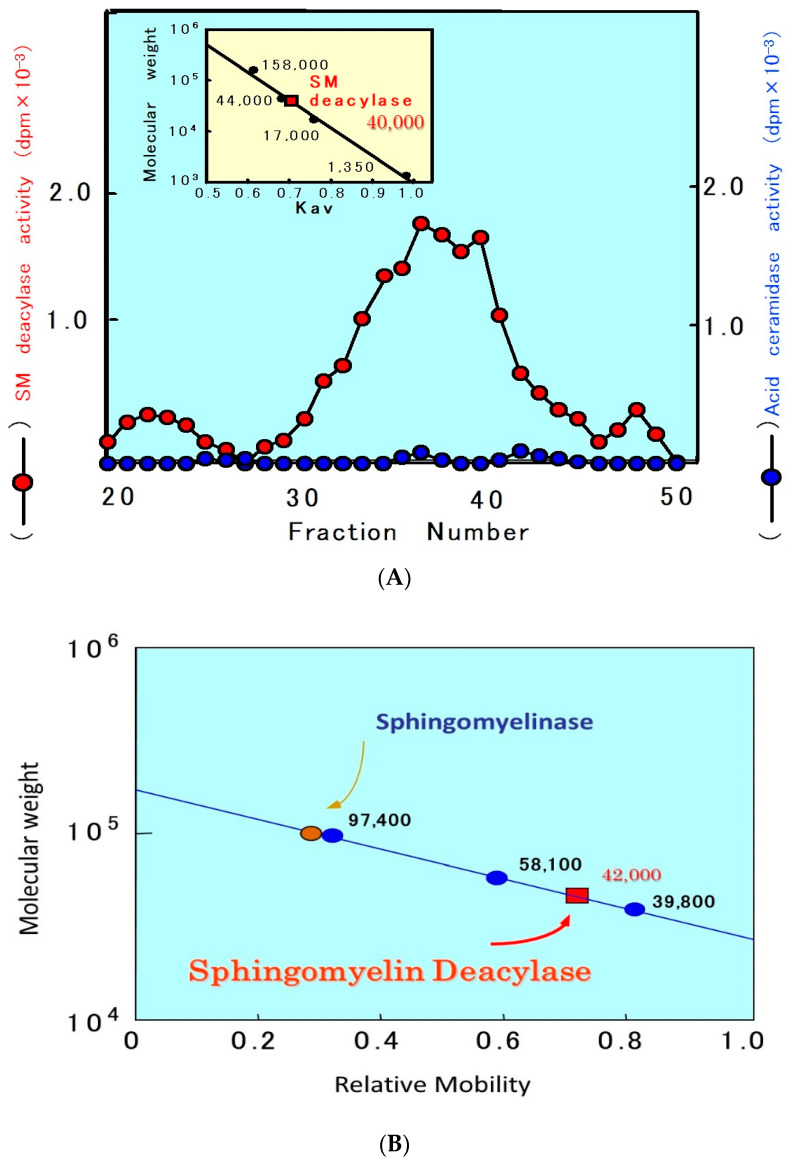
Gel chromatographic pattern of SM deacylase activity in the SC of AD skin (**A**) and the estimation of the apparent molecular mass of SM deacylase (**B**) [55].

**Figure 13 ijms-22-01613-f013:**
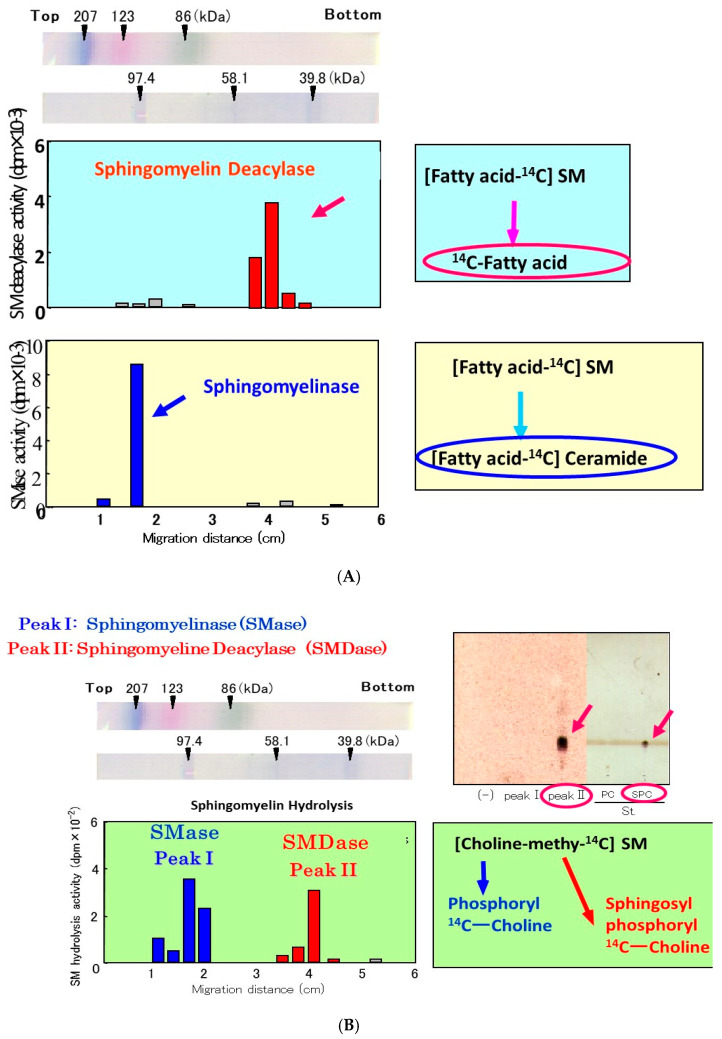
Preparative SDS-PAGE of AD epidermis for the SM hydrolysis assay using [^14^C-palmitic acid]SM (**A**) or [choline-methyl-^14^C]SM (**B**) as a substrate [55].

**Figure 14 ijms-22-01613-f014:**
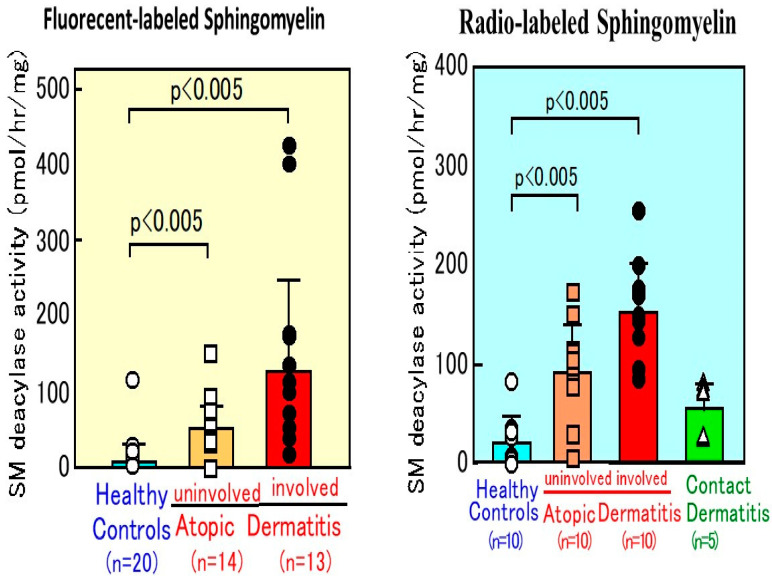
SM deacylase activity measured using fluorescent SM or radiolabeled palmitoyl SM as a substrate in the SC from AD skin or contact dermatitis skin and HC skin [50].

**Figure 15 ijms-22-01613-f015:**
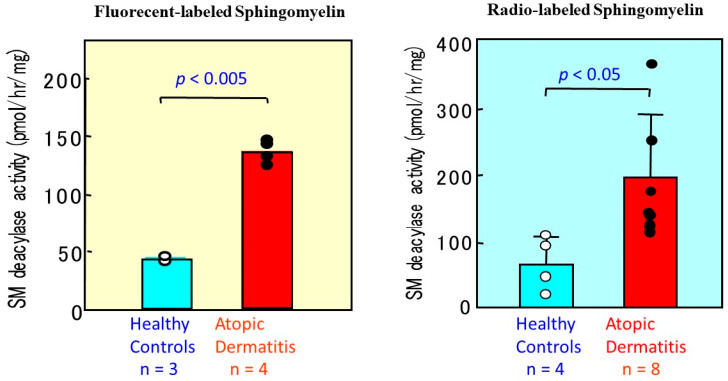
The activity of SM deacylase measured using fluorescent SM or radiolabeled [palmitoyl ^14^C]SM as a substrate in the lesional epidermis of AD skin [50].

**Figure 16 ijms-22-01613-f016:**
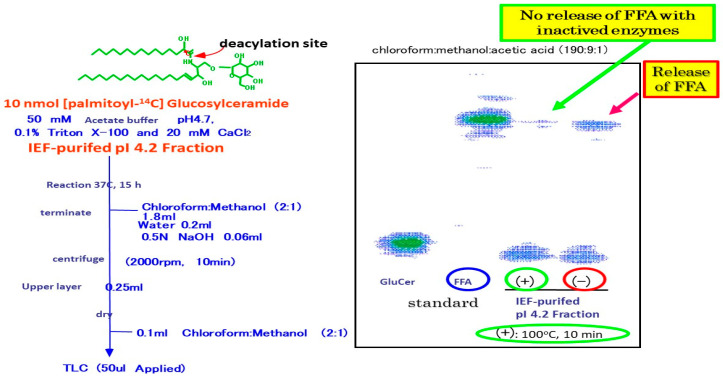
Activity of GCer deacylase by IEF-purified pI 4.2 fraction [55].

**Figure 17 ijms-22-01613-f017:**
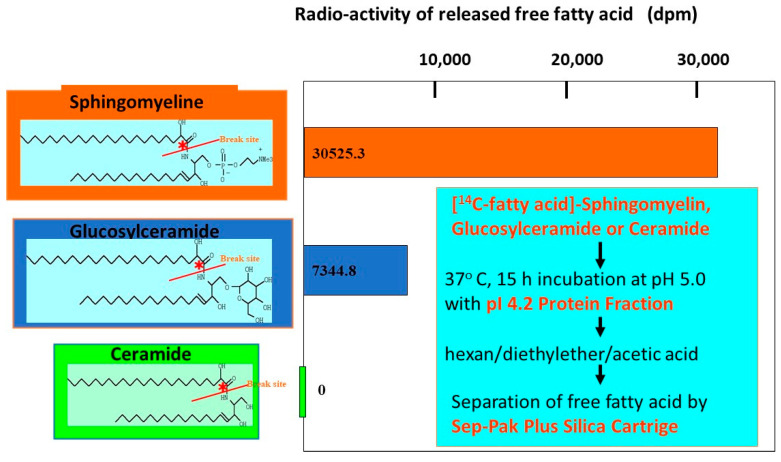
Substrate specificity of GCer SM deacylase [55].

**Figure 18 ijms-22-01613-f018:**
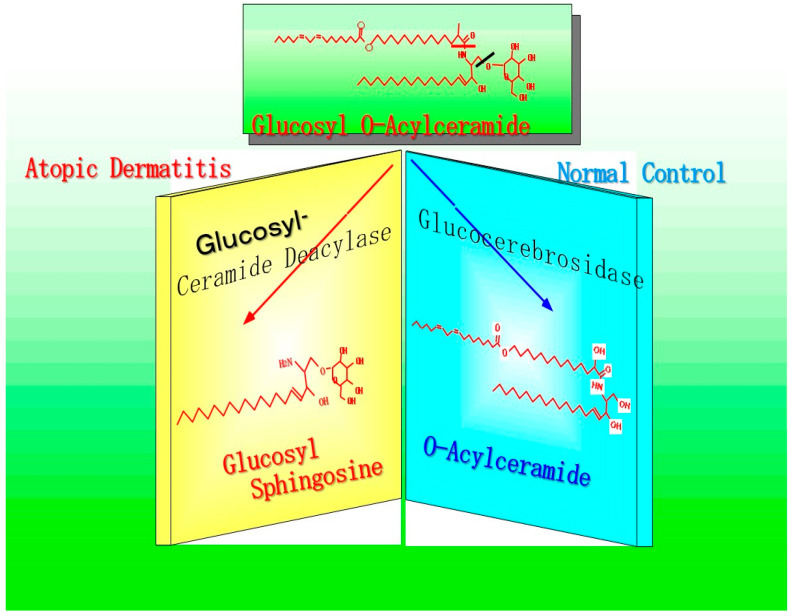
Enzymatic scheme of GCer deacylase [17].

**Figure 19 ijms-22-01613-f019:**
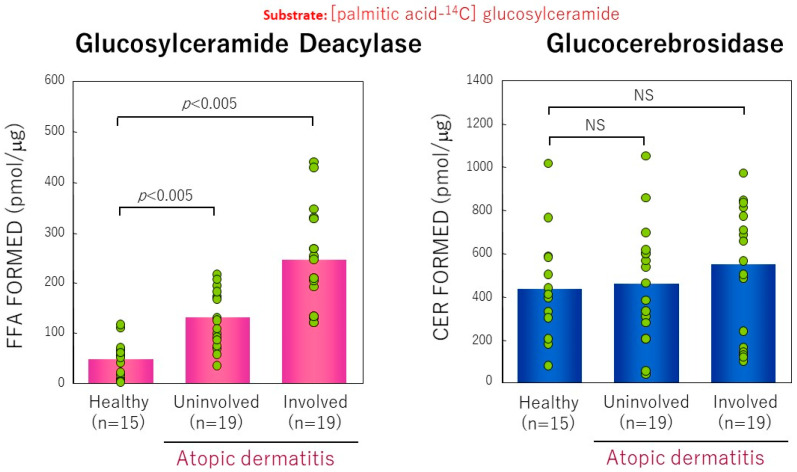
GCer deacylase activity measured using palmitoyl ^14^C-GCer as a substrate in the SC from lesional AD skin [17].

**Figure 20 ijms-22-01613-f020:**
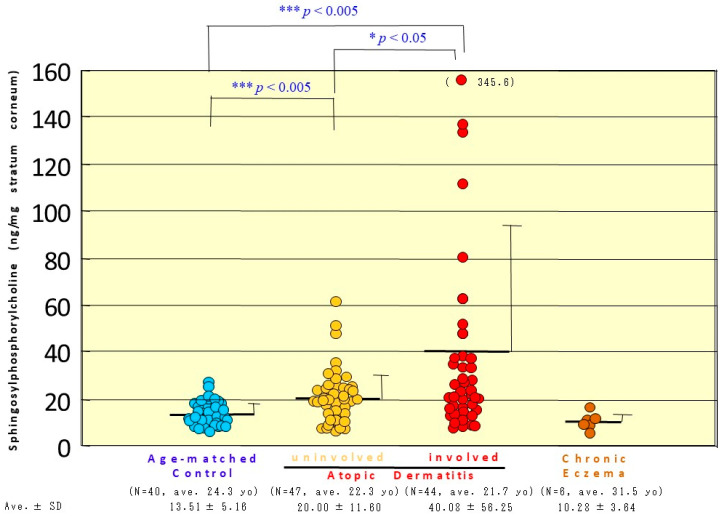
The level of SPC in the SC from the lesional or nonlesional AD skin and from chronic eczema [20].

**Figure 21 ijms-22-01613-f021:**
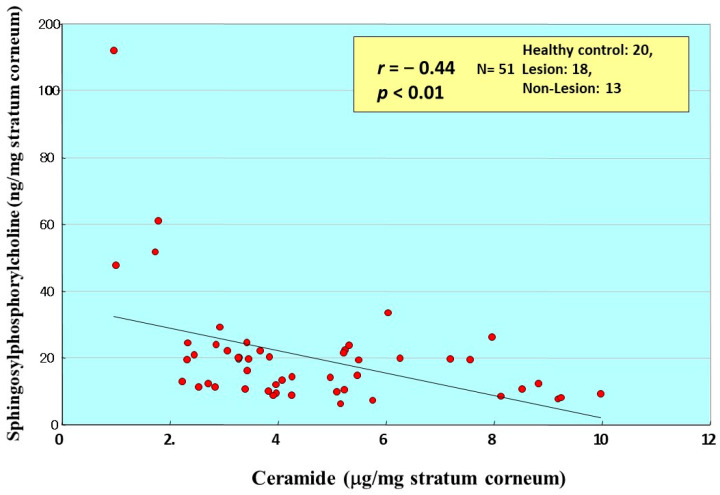
Correlation between total ceramides and SPC in the SC of AD skin [20].

**Figure 22 ijms-22-01613-f022:**
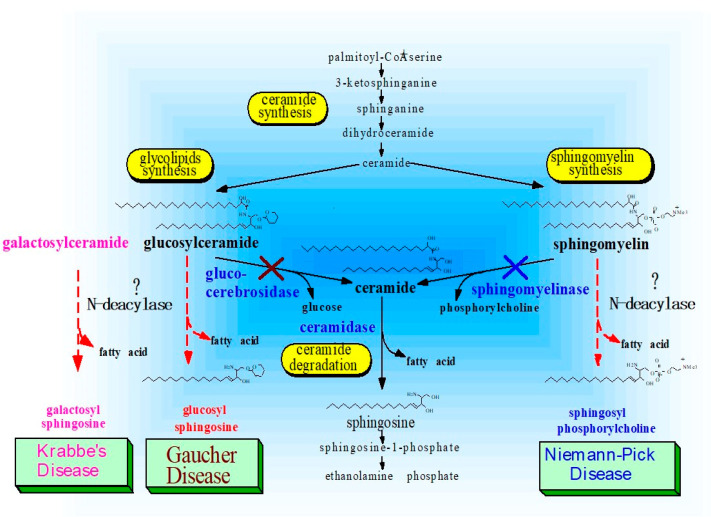
Altered sphingolipid metabolism discovered in Niemann–Pick, Gaucher and Krabbe’s Diseases [63,64,65,66,67].

**Figure 23 ijms-22-01613-f023:**
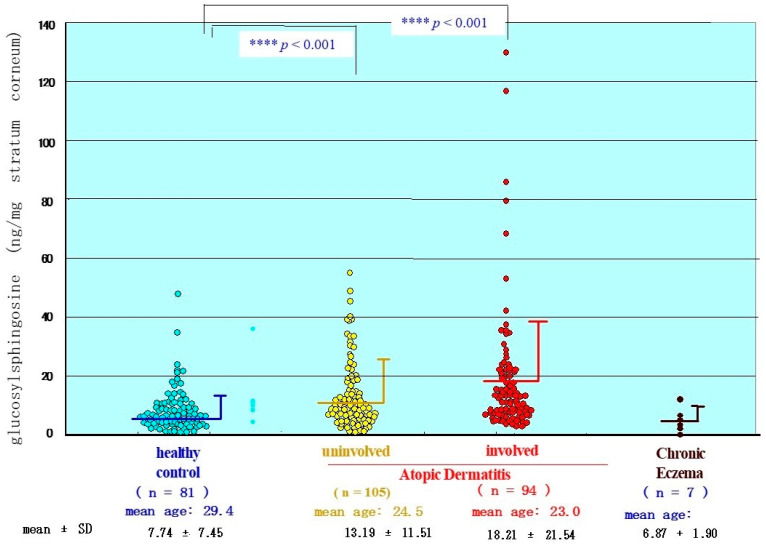
The level of GSP in the SC from the lesional or nonlesional skin with AD and chronic eczema [17].

**Figure 24 ijms-22-01613-f024:**
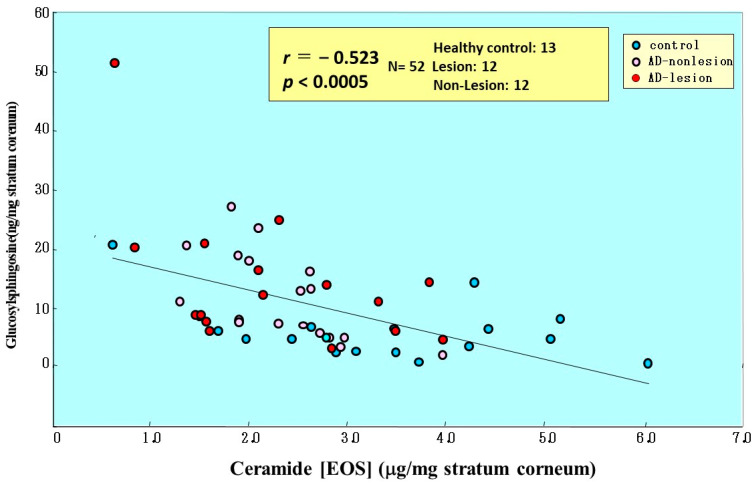
Close correlation between GSP and acylceramide (Cer[EOS]) in the SC of AD skin [17].

**Figure 25 ijms-22-01613-f025:**
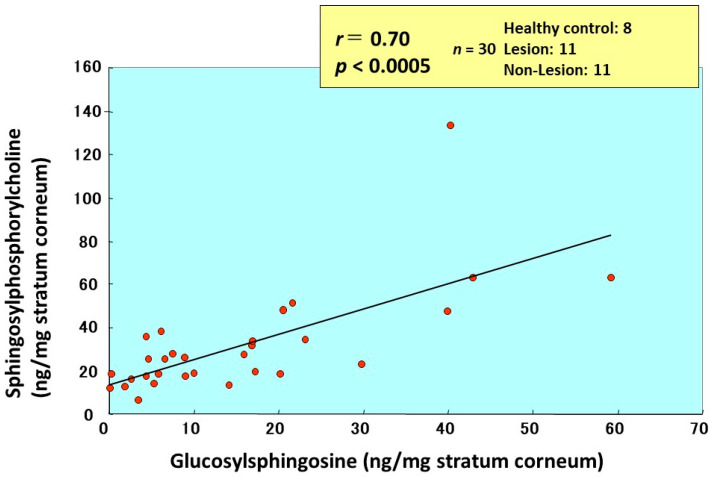
Close correlation between GSP and SPC in the SC of AD skin [17].

**Figure 26 ijms-22-01613-f026:**
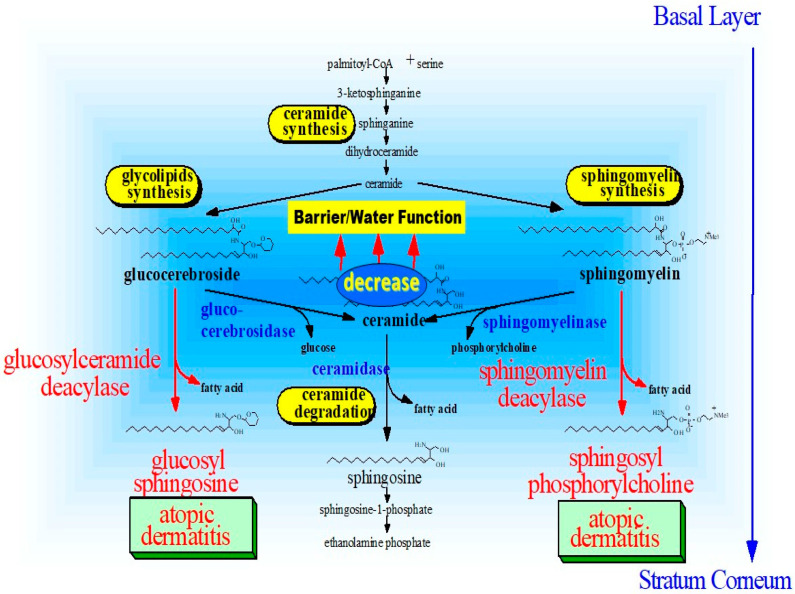
Possible biological mechanisms involved in ceramide deficiency [17,50,54,55].

**Figure 27 ijms-22-01613-f027:**
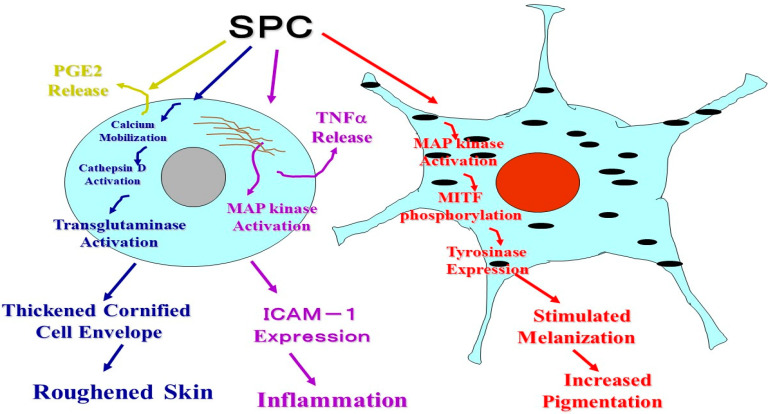
Summary of the biological effects of SPC on epidermal cells [68,69,70].

**Figure 28 ijms-22-01613-f028:**
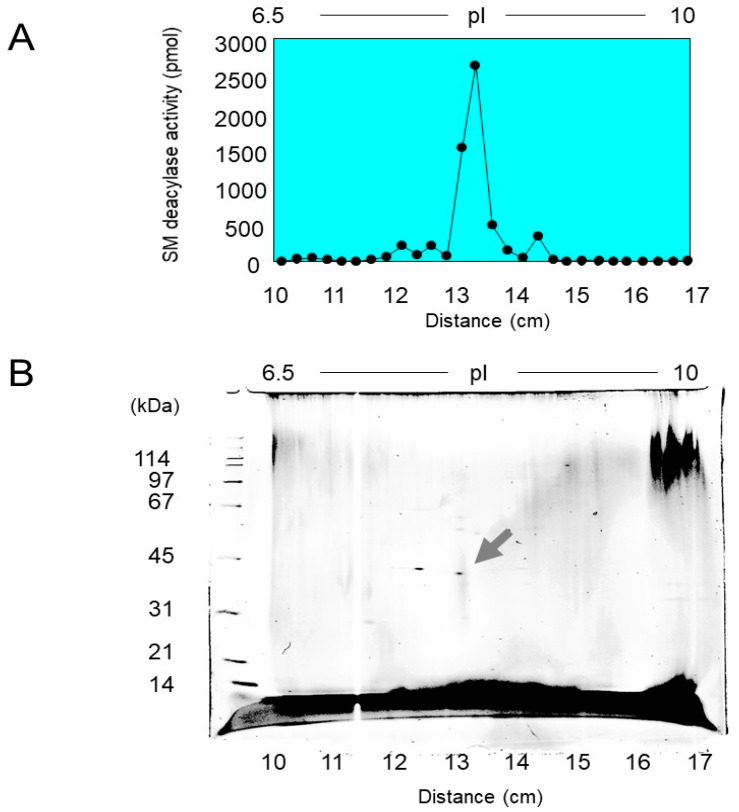
Purification and characterization of SM deacylase. (**A**) After purification by chromatography, SM deacylase was subjected to IEF, after which the IEF strips were subjected to assays for SM deacylase activity. (**B**) 2D electrophoresis was then performed by mounting an IEF separated strip gel on top of an SDS-PAGE gel. After electrophoresis, the gel was stained by Cypro-Ruby and detected using a fluorescence image scanner. The protein spot indicated by the arrow was subjected to MS/MS analysis [71].

**Figure 29 ijms-22-01613-f029:**
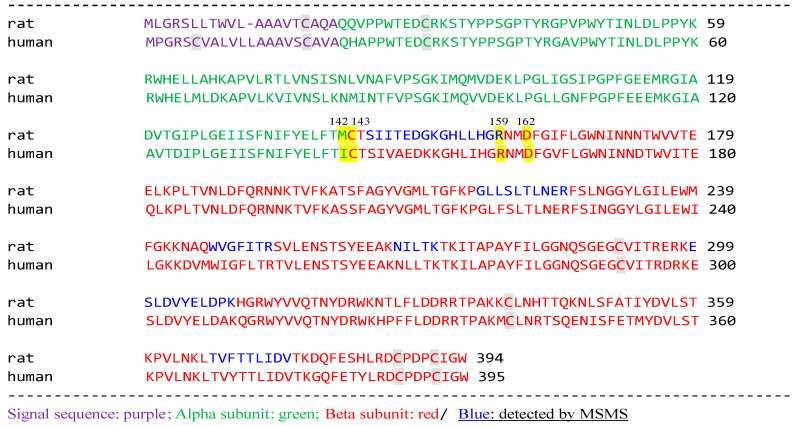
aCDase β-subunit hits by MASCOT database [71].

**Figure 30 ijms-22-01613-f030:**
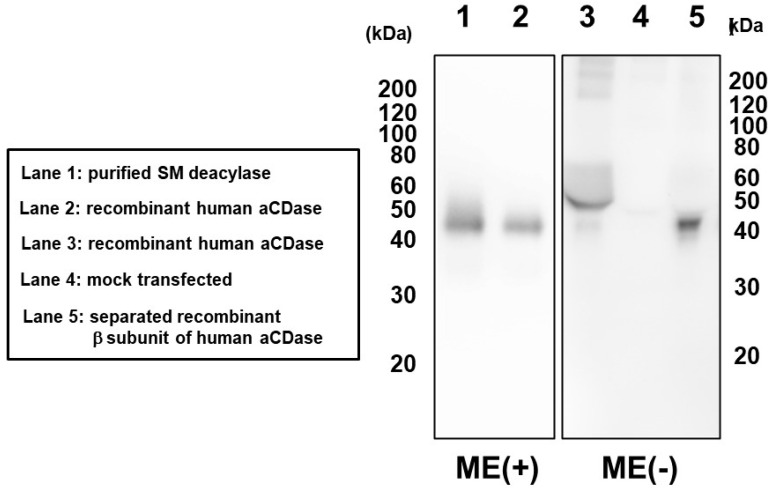
Subunit composition of purified SM deacylase recombinant human aCDase and separated recombinant β-subunit of human aCDase. The samples were separated by SDS-PAGE followed by immunoblot analysis using antibodies to the β-subunit (human) of aCDase. Before electrophoresis, samples were reduced with 5% 2-mercaptoethanol (ME) indicated by ME+ but were not treated with ME indicated by ME-. Lane 1, purified rat SM deacylase (ME+); Lane 2, recombinant human aCDase (ME+); Lane 3, recombinant human aCDase (ME-). Lane 4, mock transfected (ME-); Lane 5, separated recombinant β-subunit of human aCDase (ME-) [71].

**Figure 31 ijms-22-01613-f031:**
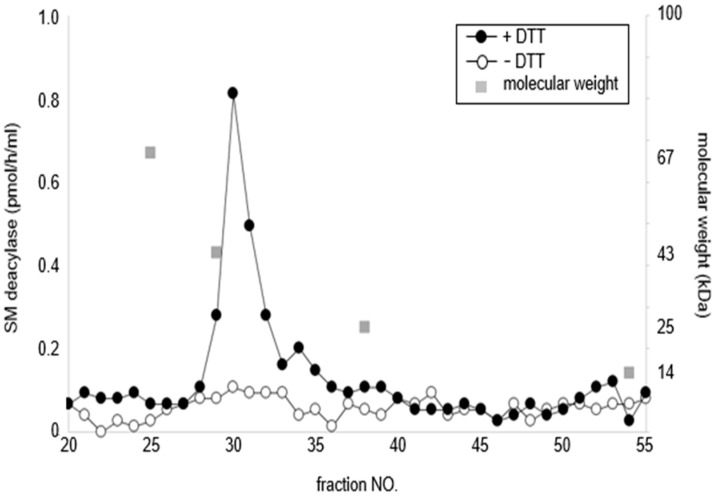
Treatment with DTT separates SM deacylase from recombinant human aCDase. Recombinant human aCDase was incubated in a buffer for 60 min with (solid circles) or without (open circles) DTT at 200 mM and was then subjected to gel filtration chromatography using a Superdex 200 column. Proteins were eluted and fractions were collected then analyzed for activities of SM deacylase [71].

**Figure 32 ijms-22-01613-f032:**
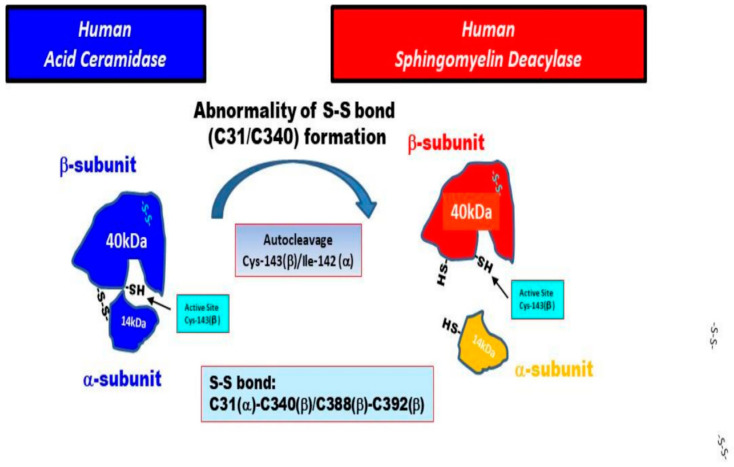
Hypothetical mechanisms involved in the expression of SM deacylase in AD skin [71].

**Table 1 ijms-22-01613-t001:** Correlation coefficients between ceramide classes (ng/μg protein)/penetrated levels of pCer (ng/μg protein) and scheme 13. *N* = 39. ** *p* < 0.01.

Endogenous Ceramide Species & pCer	Skin Conductance (µS)		Trans-Epidermal Water Loss (g/m^2^/h)	
	Correlation Coefficient	*p*-Value	Correlation Coefficient	*p*-Value
pCer	0.436	0.0056 **	0.1015	0.5386
Total Ceramides	−0.2372	0.1459	-0.0769	0.6417
Cer [NDS]	0.2357	0.1478	-0.1312	0.4259
Cer [NS]	−0.2141	0.1905	0.1739	0.2898
Cer [NH]	−0.2049	0.2108	0.0157	0.9242
Cer [NP]	−0.1506	0.36	−0.1586	0.335
Cer [ADS]	−0.2069	0.2604	−0.2302	0.1585
Cer [AS]	−0.2399	0.1413	0.0867	0.5992
Cer [AH]	−0.2054	0.2097	0.0017	0.9917
Cer [AP]	−0.2609	0.1087	−0.2694	0.0972
Cer [EOS]	−0.1978	0.2275	0.1239	0.42
Cer [EOH]	−0.2045	0.2117	0.0301	0.8559
Cer [EOP]	−0.197	0.2292	−0.1329	0.4119

## Data Availability

Not applicable.

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
