# Peer review of "Cutting Edge of the Pathogenesis of Atopic Dermatitis: Sphingomyelin Deacylase, the Enzyme Involved in Its Ceramide Deficiency, Plays a Pivotal Role"

_ijms, 2021, doi:10.3390/ijms22041613_

Round 1
Reviewer 1 Report
The article is an interesting review on the role of Sphingomyelin Deacylase in the development of Atopic Dermatitis. The paper is basically a narrative and complete revision of all the literature on the topic, highlighting how this enzyme downregulates ceramide production in Atopic dermatitis and suggesting it as a potential target for future drugs.
The main problem with this article, and also the reason for which a major revision is required, is the lack of a materials and methods section highlighting the modalities and databases of the selection of the studies.
Also, in the introduction, a small paragraph describing the condition clinically and the possible treatments present nowadays should be added; Here I suggest two interesting articles: "Nettis E, Ortoncelli M, Pellacani G et al. A Multicenter Study on the Prevalence of Clinical Patterns and Clinical Phenotypes in Adult Atopic Dermatitis. J Investig Allergol Clin Immunol. 2020;30(6):448-450." and "Dattola A, Bennardo L, Silvestri M, Nisticò SP. What's new in the treatment of atopic dermatitis? Dermatol Ther. 2019 Mar;32(2):e12787."
Thank You
Author Response
Reviewer 1:
The article is an interesting review on the role of Sphingomyelin Deacylase in the development of Atopic Dermatitis. The paper is basically a narrative and complete revision of all the literature on the topic, highlighting how this enzyme downregulates ceramide production in Atopic dermatitis and suggesting it as a potential target for future drugs.
The main problem with this article, and also the reason for which a major revision is required, is the lack of a materials and methods section highlighting the modalities and databases of the selection of the studies.
Response:
Because this manuscript is a review article, I did not describe things related to Materials and Method section.
--------------------------------------------------------------------------
Also, in the introduction, a small paragraph describing the condition clinically and the possible treatments present nowadays should be added; Here I suggest two interesting articles: "Nettis E, Ortoncelli M, Pellacani G et al. A Multicenter Study on the Prevalence of Clinical Patterns and Clinical Phenotypes in Adult Atopic Dermatitis. J Investig Allergol Clin Immunol. 2020;30(6):448-450." and "Dattola A, Bennardo L, Silvestri M, Nisticò SP. What's new in the treatment of atopic dermatitis? Dermatol Ther. 2019 Mar;32(2):e12787."
Response:
Considering the reviewer’s comments, I added ref No1 in the first page and ref No 82 in the Conclusion as well as in the References as marked by track changes in the revised manuscript with track changes.
----------------------------------------------------------------------
Reviewer 2 Report
Abstract and paper very long .Can be shortened
Author Response
Reviewer 2:
Abstract and paper very long . Can be shortened
Response:
Considering the reviewer’s comments, I have shortened the abstract and the text as much as possible as marked by track changes in the revised manuscript with track changes.
Reviewer 3 Report
Brief Summary:
The concept of summarizing literature information about Sphingomyelin Deacylase and its role in Atopic Dermatitis is very interesting and of high importance, especially for the researchers working on the field of Atopic Dermatitis. It highly meets the criteria to be included in the special issue entitled Molecular Mechanisms of Skin Aging and Atopic Dermatitis. Obviously, the best author to do so, is the author of this review whose research group contributed to a predominant degree in discovering and identifying the properties of the enzyme. It is noteworthy that 47 out of 115 references are from the author’s group.
Broad comments:
The manuscript is written with a very nice order of paragraphs covering all the major findings about Sphingomyelin Deacylase up to date. I would also like to highlight the usage of questioning titles as an advantage for the scope of the manuscript. On the other hand, the manuscript has two major disadvantages. Initially, the text contains very long sentences that are for the reader difficult to follow. This drawback is more intense in the first pages of the manuscript. I would highly recommend an average of 25-30 words/sentence through the text. In addition, there is sometimes extended analysis of previously published experimental data (accompanied with noticeable text similarities) from the author’s group in the manuscript. The extensive use of the author’s research work is understandable and logical in this manuscript. On the other hand, this is a fact which is reducing the manuscript’s prestige and jeopardizing its potentially beneficial impact on the research community. I would recommend extensive revision with a drastic reduction of the text size by avoiding (as much as possible) the repetition of previously published text. Instead, I would recommend the text to be more direct with the findings and enriched with the expert author’s comments on the long-term collected findings.
Specific Comments:
- Abstract:
- Too long text for an Abstract. I would recommend shortening it.
- Emphasis on group’s results.
- Long sentences in the abstract. Re-reading is usually needed to get the exact meaning. I would suggest limiting the words per sentence to an average of 25-30.
- aCDase abbreviation is used without explanation.
- Missing word at the last sentence of the abstract “…as the β-subunit of aCDase that…”.
- 1st part; Water-loss and AD connection:
- Several articles are mentioned but only the group’s paper (ref 11) is commented in detail.
- Long sentences.
- 2nd part; abnormality in Percutaneous Permeability:
- No other references except the group’s work (ref 2). I would suggest the review Br J Dermatol 2017, 177, 1, 84-106 (DOI: 10.1111/bjd.15065) and the references therein.
- Word change at the text “during the topical application period for 2 hr” to “during topical application period of 2 hr”.
- 3rd part; Is barrier disruption the cause or the result?:
- The same group’s articles referred (2,11).
- A not clear statement is given as an answer. It seems it is a non-stopping cycle. Please clarify better.
- Long sentences.
- 4th part; Ceramide deficiency:
- “The essential… …SC layer.” is a very long sentence.
- 5th part; Significance of Cer in SC functions.
- Extended reference to group’s article (ref 31) about pCer.
- 6th part; Significance of the Cer profile in SC functions:
- Nicely given outcome.
- 7th part; Inherent barrier disruption or not?
- The outcome for the given question (barrier disruption not inherent) is given very early in the paragraph based on other studies and the group’s studies.
- Emphasis on group’s reference.
- 8th part; Clinical evidence of the damaged Cer-production process:
- Emphasis on group’s article (ref 20).
- Extended experimental data explanation.
- The use of acid sphingomyelinase in the main text for the first time. The abbreviation should be added.
- Overuse of reference 51 (page 10). It should be added at the end of the sentence “…in uninflamed non-lesional AD skin.”
- “Because there is… …level of BGCase.” is a very long sentence.
- 9th part; Discovery of Sphingomyelin Deacylase:
- The whole paragraph has text similarities with another paragraph from Review J. Dermatol. Sci 2009, 55, 1-9. (DOI: https://doi.org/10.1016/j.jdermsci.2009.04.006).
- The sentence “Based on the TLC… …deacylase-like enzyme.” should be formatted together with the text, not with the figure.
- Page 14, There are two almost identical sentences: In the assay for SM hydrolysis with N-[palmitoyl-1-14C]SM (Figure 12A), there were two separated peaks of radioactivity distributed in the lipophilic phase, each corresponding in electrophoretic mobility to the molecular weights estimated for SM deacylase and In the assay for SM hydrolysis with [choline-methyl14C]SM (Figure 12B), there were two separated peaks (peaks I and II) of radioactivity distributed in the aqueous phase, each corresponding in electrophoretic mobility to the molecular weights estimated for SM deacylase and aSMase. I would suggest re-writing to be more friendly to the reader.
- 10th part; SM Deacylase activity:
- Text similarities with reference 49.
- Example 1:
- Original text from reference 49: In assays to measure the release of [1±14C] palmitic acid as SM deacylase activity, the released amounts of palmitic acid would not necessarily represent the activity of SM deacylase if both acid SMase and CDase are present in stratum corneum extracts from AD patients. This is due to the fact that the release of radiolabeled palmitic acid could also result from a two step reaction where SM would be initially hydrolyzed by acid SMase to yield radiolabeled palmitoyl sphingosine (ceramide), which in turn could be converted by acid CDase to radiolabeled palmitic acid. The negligible levels of acid CDase in AD stratum corneum, however, allow us to conclude that the released amounts of radiolabeled palmitic acid reflect direct cleavage of [palmitic acid-14C] SM at the acyl site
- Text from manuscript page 17: In assays using radiolabeled [palmitic acid14C]SM as a substrate, the amount of [1-14C]palmitic acid released does not necessarily represent the activity of SM deacylase if both aSMase and aCDase are present in the SC extract [49]. This is because the release of radiolabeled palmitic acid could also result from a two-step reaction wherein SM would be hydrolyzed by aSMase to yield radiolabeled palmitoylsphingosine (ceramide) which in turn could be converted by aCDase to radiolabeled palmitic acid. However, the negligible levels of aCDase in the SC of patients with AD [49] allowed us to conclude that the amount of radiolabeled palmitic acid released reflects the direct cleavage of [palmitic acid14C]SM at the acyl site.
- Example 2:
- Original text from reference 49: This study has demonstrated that SM deacylase activity is enhanced more than 5-fold in involved stratum corneum and more than 3-fold in uninvolved stratum corneum of AD patients, compared with normal stratum corneum. In contrast, the stratum corneum from patients with contact dermatitis showed no increases in SM deacylase activity compared with healthy controls, suggesting that changes in SM deacylase activity are unlikely to be involved in the general etiology of cutaneous inflammation. Our previous study on the epidermal localization of CDase and GCase, the latter being a hydrolytic enzyme in intercellular spaces between the stratum corneum and the granular layer, suggested that the activities of ceramide metabolism-related enzymes within the stratum corneum approximately represent the epidermal activities of the same enzymes (Holleranet al, 1992; Yadaet al, 1995; Takagiet al,1999). Consistent with this relationship, similar high levels of SMdeacylase activity were detected in the epidermis from AD patients, whereas there was no significant difference in SMase between AD and healthy controls, suggesting that epidermal cells from these patients show abnormal production of a hitherto undiscovered epidermal enzyme, termed SM deacylase. To exclude the possibility that the SM deacylase activity observed was due to contamination by bacteria, such as Staphylococcus aureus, which are often present on the surface of the stratum corneum of AD patients(Leydenet al, 1974; Dahl, 1983), we examined whether high levels of SM deacylase activity were present in epidermis freed of bacteria by excessive tape-stripping of the stratum corneum before biopsy. The SM deacylase activities of such tape-stripped skin still showed higher levels in AD patients, suggesting that the high SM deacylase activity noted is not derived from bacterial contamination of the stratum corneum.
- Text from review page 17: As shown in Figure 13, our quantitative measurements [49] clearly demonstrated that SM deacylase activity is enhanced more than 5-fold in lesional SC and more than 3- fold in non-lesional SC of AD skin, compared with the SC of HC skin. In contrast, the SC from patients with contact dermatitis showed no increase in SM deacylase activity compared with HCs, which suggests that changes in SM deacylase activity are unlikely to be involved in the general etiology of cutaneous inflammation. Our earlier study on the epidermal localization of aCDase and BGCase, the latter being a hydrolytic enzyme localized in intercellular spaces between the SC and the granular layer, suggested that the activities of ceramide metabolism-related enzymes within the SC approximately represent the epidermal activities of the same enzymes [55, 56, 57]. Consistent with that relationship, a similar higher level of SM deacylase activity was detected in the epidermis from AD skin, whereas there was no significant difference in levels of aSMase between AD skin and HC skin (Figure 14) [49], which suggests that epidermal cells from AD patients show abnormal production of the hitherto undiscovered epidermal enzyme termed SM deacylase. To exclude the possibility that the SM deacylase activity observed was due to contamination by bacteria, such as Staphylococcus aureus, which are often present on the surface of AD skin [58, 59], we examined whether high levels of SM deacylase activity were present in epidermis freed of bacteria by excessive tape-stripping of the SC before biopsy. It turned out that the SM deacylase activities of such tape-stripped skin is still higher than is found in AD skin [49], which suggests that the high SM deacylase activity noted is not derived from bacterial contamination of the SC.
- Example 1:
- Text similarities with reference 49.
- 11th part, GCer deacylase activity in AD Skin:
- Text similarities with J. Dermatol. Sci 2009, 55, 1-9. (DOI:https://doi.org/10.1016/j.jdermsci.2009.04.006).
- 12th part, Accumulation of SPC:
- The parts „Quantitive analysis… …with AD compared to HC skin.” and „Since increased… …in the SC of AD skin” contain similarities with the text from reference 19.
- 13th part, Accumulation of GSP:
- The part „Quantitative analysis of GSP… …compared to age-matched HC skin.” contains similarities with the text from reference 16.
- Page 26, at the end: SM deacylase is extensively explained, even though it was explained earlier in the text.
- Word change suggestion, page 28, at the end: “Interestingly, the enzymatic reaction products, SPC and GSP, which are essential surrogates to determine whether SM GCer deacylase is functioning in situ in the epidermis, are significantly increased in the non-lesional and lesional SC of patients with AD compared with HCs, and are reciprocally related to the decreased levels of ceramides in a similar group of patients with AD.”
- Page 29, Figure 25: β-glucocerebrosidase instead of glucocerebrosidase.
- 14th part, SPC Stimulates ICAM-1 Expression in Human Keratinocytes:
- The part „The expression of ICAM-1… …to augment the ICAM-1 expression.” contains similarities with the text from reference 16.
- 15th part, SPC Stimulates TGase in Human Keratinocytes:
- Similarities with reference 83.
- 16th part, SPC Stimulates Melanogenesis in Human Melanocytes:
- Similarities with reference 91.
- 18th part, Purification of SM Deacylase:
- Similarities with reference 106.
- 19th part, Enzymatic properties of Purified SM Deacylase:
- Similarities with reference 106.
- 20th part, Identification of SM Deacylase at the Protein Level:
- Similarities with reference 106.
Author Response
Reviewer 3:
Brief Summary:
The concept of summarizing literature information about Sphingomyelin Deacylase and its role in Atopic Dermatitis is very interesting and of high importance, especially for the researchers working on the field of Atopic Dermatitis. It highly meets the criteria to be included in the special issue entitled Molecular Mechanisms of Skin Aging and Atopic Dermatitis. Obviously, the best author to do so, is the author of this review whose research group contributed to a predominant degree in discovering and identifying the properties of the enzyme. It is noteworthy that 47 out of 115 references are from the author’s group.
Broad comments:
The manuscript is written with a very nice order of paragraphs covering all the major findings about Sphingomyelin Deacylase up to date. I would also like to highlight the usage of questioning titles as an advantage for the scope of the manuscript. On the other hand, the manuscript has two major disadvantages. Initially, the text contains very long sentences that are for the reader difficult to follow. This drawback is more intense in the first pages of the manuscript. I would highly recommend an average of 25-30 words/sentence through the text. In addition, there is sometimes extended analysis of previously published experimental data (accompanied with noticeable text similarities) from the author’s group in the manuscript. The extensive use of the author’s research work is understandable and logical in this manuscript. On the other hand, this is a fact which is reducing the manuscript’s prestige and jeopardizing its potentially beneficial impact on the research community. I would recommend extensive revision with a drastic reduction of the text size by avoiding (as much as possible) the repetition of previously published text. Instead, I would recommend the text to be more direct with the findings and enriched with the expert author’s comments on the long-term collected findings.
Response:
Considering the reviewer’s broad comments, I have shortened the abstract and the text as much as possible as marked by track changes in the revised manuscript with track changes.
------------------------------------------------------------------------------------
Specific Comments:
Abstract:
Too long text for an Abstract. I would recommend shortening it.
Emphasis on group’s results.
Long sentences in the abstract. Re-reading is usually needed to get the exact meaning. I would suggest limiting the words per sentence to an average of 25-30.
aCDase abbreviation is used without explanation.
Missing word at the last sentence of the abstract “…as the β-subunit of aCDase that…”.
Response:
Considering the reviewer’s comments, I have modified the abstract by reducing several sentences and added acid ceramidase (aCDase) as marked by track changes in the Abstract of the revised manuscript with track changes.
----------------------------------------------------------------------
1st part; Water-loss and AD connection:
Several articles are mentioned but only the group’s paper (ref 11) is commented in detail.
Long sentences.
Response:
Because among the related papers, ref 11 was the most suitable paper, we commented it in detail as a representative information.
----------------------------------------------------------------
2nd part; abnormality in Percutaneous Permeability:
No other references except the group’s work (ref 2). I would suggest the review Br J Dermatol 2017, 177, 1, 84-106 (DOI: 10.1111/bjd.15065) and the references therein.
Word change at the text “during the topical application period for 2 hr” to “during topical application period of 2 hr”.
Response:
According to the reviewer’s suggestion, I have added the review Br J Dermatol 2017, 177, 1, 84-106 (DOI: 10.1111/bjd.15065) as ref No 14 in the text and in the Reference section as marked by track changes in the revised manuscript with remarks.
I also corrected the indicated sentences as marked by track changes in the revised manuscript with remarks.
---------------------------------------------------------------------
3rd part; Is barrier disruption the cause or the result?:
The same group’s articles referred (2,11).
A not clear statement is given as an answer. It seems it is a non-stopping cycle. Please clarify better.
Long sentences.
Response:
Considering the reviewer’s comments, I modified the related sentences by deleting some and by adding some sentences as marked by track changes in the revised manuscript with track changes.
----------------------------------------------------------------------
4th part; Ceramide deficiency:
“The essential… …SC layer.” is a very long sentence.
Response:
Considering the reviewer’s comments, I modified the related sentences by deleting some as marked by track changes in the revised manuscript with track changes.
------------------------------------------------------------------------------
5th part; Significance of Cer in SC functions.
Extended reference to group’s article (ref 31) about pCer.
6th part; Significance of the Cer profile in SC functions:
Nicely given outcome.
7th part; Inherent barrier disruption or not?
The outcome for the given question (barrier disruption not inherent) is given very early in the paragraph based on other studies and the group’s studies.
Emphasis on group’s reference.
8th part; Clinical evidence of the damaged Cer-production process:
Emphasis on group’s article (ref 20).
Extended experimental data explanation.
The use of acid sphingomyelinase in the main text for the first time. The abbreviation should be added.
Overuse of reference 51 (page 10). It should be added at the end of the sentence “…in uninflamed non-lesional AD skin.”
“Because there is… …level of BGCase.” is a very long sentence.
Response:
Considering the reviewer’s comments, I changed the site of ref 51 and modified the related sentences to reduce the length of sentences some as marked by track changes in the revised manuscript with track changes.
--------------------------------------------------------------
9th part; Discovery of Sphingomyelin Deacylase:
The whole paragraph has text similarities with another paragraph from Review J. Dermatol. Sci 2009, 55, 1-9. (DOI: https://doi.org/10.1016/j.jdermsci.2009.04.006).
The sentence “Based on the TLC… …deacylase-like enzyme.” should be formatted together with the text, not with the figure.
Response:
The pointed formatted era was corrected.
Considering the reviewer’s comments about the similarities, in order to reduce the similarity as much as possible, I have extensively modified many sentences by deleting many sentences and by rewriting many sentences as marked by track changes in the revised manuscript with track changes.
--------------------------------------------------------------------
Page 14, There are two almost identical sentences: In the assay for SM hydrolysis with N-[palmitoyl-1-14C]SM (Figure 12A), there were two separated peaks of radioactivity distributed in the lipophilic phase, each corresponding in electrophoretic mobility to the molecular weights estimated for SM deacylase and In the assay for SM hydrolysis with [choline-methyl14C]SM (Figure 12B), there were two separated peaks (peaks I and II) of radioactivity distributed in the aqueous phase, each corresponding in electrophoretic mobility to the molecular weights estimated for SM deacylase and aSMase. I would suggest re-writing to be more friendly to the reader.
Response:
According to the reviewer’s suggestions, I rewrote the indicated sentences as marked by track changes in the revised manuscript with track changes.
-----------------------------------------------------------------------------------
10th part; SM Deacylase activity:
Text similarities with reference 49.
Example 1:
Original text from reference 49: In assays to measure the release of [1±14C] palmitic acid as SM deacylase activity, the released amounts of palmitic acid would not necessarily represent the activity of SM deacylase if both acid SMase and CDase are present in stratum corneum extracts from AD patients. This is due to the fact that the release of radiolabeled palmitic acid could also result from a two step reaction where SM would be initially hydrolyzed by acid SMase to yield radiolabeled palmitoyl sphingosine (ceramide), which in turn could be converted by acid CDase to radiolabeled palmitic acid. The negligible levels of acid CDase in AD stratum corneum, however, allow us to conclude that the released amounts of radiolabeled palmitic acid reflect direct cleavage of [palmitic acid-14C] SM at the acyl site
Text from manuscript page 17: In assays using radiolabeled [palmitic acid14C]SM as a substrate, the amount of [1-14C]palmitic acid released does not necessarily represent the activity of SM deacylase if both aSMase and aCDase are present in the SC extract [49]. This is because the release of radiolabeled palmitic acid could also result from a two-step reaction wherein SM would be hydrolyzed by aSMase to yield radiolabeled palmitoylsphingosine (ceramide) which in turn could be converted by aCDase to radiolabeled palmitic acid. However, the negligible levels of aCDase in the SC of patients with AD [49] allowed us to conclude that the amount of radiolabeled palmitic acid released reflects the direct cleavage of [palmitic acid14C]SM at the acyl site.
Example 2:
Original text from reference 49: This study has demonstrated that SM deacylase activity is enhanced more than 5-fold in involved stratum corneum and more than 3-fold in uninvolved stratum corneum of AD patients, compared with normal stratum corneum. In contrast, the stratum corneum from patients with contact dermatitis showed no increases in SM deacylase activity compared with healthy controls, suggesting that changes in SM deacylase activity are unlikely to be involved in the general etiology of cutaneous inflammation. Our previous study on the epidermal localization of CDase and GCase, the latter being a hydrolytic enzyme in intercellular spaces between the stratum corneum and the granular layer, suggested that the activities of ceramide metabolism-related enzymes within the stratum corneum approximately represent the epidermal activities of the same enzymes (Holleranet al, 1992; Yadaet al, 1995; Takagiet al,1999). Consistent with this relationship, similar high levels of SMdeacylase activity were detected in the epidermis from AD patients, whereas there was no significant difference in SMase between AD and healthy controls, suggesting that epidermal cells from these patients show abnormal production of a hitherto undiscovered epidermal enzyme, termed SM deacylase. To exclude the possibility that the SM deacylase activity observed was due to contamination by bacteria, such as Staphylococcus aureus, which are often present on the surface of the stratum corneum of AD patients(Leydenet al, 1974; Dahl, 1983), we examined whether high levels of SM deacylase activity were present in epidermis freed of bacteria by excessive tape-stripping of the stratum corneum before biopsy. The SM deacylase activities of such tape-stripped skin still showed higher levels in AD patients, suggesting that the high SM deacylase activity noted is not derived from bacterial contamination of the stratum corneum.
Text from review page 17: As shown in Figure 13, our quantitative measurements [49] clearly demonstrated that SM deacylase activity is enhanced more than 5-fold in lesional SC and more than 3- fold in non-lesional SC of AD skin, compared with the SC of HC skin. In contrast, the SC from patients with contact dermatitis showed no increase in SM deacylase activity compared with HCs, which suggests that changes in SM deacylase activity are unlikely to be involved in the general etiology of cutaneous inflammation. Our earlier study on the epidermal localization of aCDase and BGCase, the latter being a hydrolytic enzyme localized in intercellular spaces between the SC and the granular layer, suggested that the activities of ceramide metabolism-related enzymes within the SC approximately represent the epidermal activities of the same enzymes [55, 56, 57]. Consistent with that relationship, a similar higher level of SM deacylase activity was detected in the epidermis from AD skin, whereas there was no significant difference in levels of aSMase between AD skin and HC skin (Figure 14) [49], which suggests that epidermal cells from AD patients show abnormal production of the hitherto undiscovered epidermal enzyme termed SM deacylase. To exclude the possibility that the SM deacylase activity observed was due to contamination by bacteria, such as Staphylococcus aureus, which are often present on the surface of AD skin [58, 59], we examined whether high levels of SM deacylase activity were present in epidermis freed of bacteria by excessive tape-stripping of the SC before biopsy. It turned out that the SM deacylase activities of such tape-stripped skin is still higher than is found in AD skin [49], which suggests that the high SM deacylase activity noted is not derived from bacterial contamination of the SC.
Response:
There is no choice for completely deleting the similarity because this review article focuses only on our own published papers related to atopy pathogenesis. However, considering the reviewer’s comments about the similarities, in order to reduce the similarity as much as possible, I have extensively modified many sentences by deleting many sentences and by rewriting many sentences as marked by track changes in the revised manuscript with track changes.
-------------------------------------------------------------------------------------
11th part, GCer deacylase activity in AD Skin:
Text similarities with J. Dermatol. Sci 2009, 55, 1-9. (DOI:https://doi.org/10.1016/j.jdermsci.2009.04.006).
Response:
There is no choice for completely deleting the similarity because this review article focuses on only our own published papers related to atopy pathogenesis. However, Considering the reviewer’s comments about the similarities, in order to reduce the similarity as much as possible, I have extensively modified many sentences by deleting many sentences and by rewriting many sentences as marked by track changes in the revised manuscript with track changes.
-------------------------------------------------------------------------------------
12th part, Accumulation of SPC:
The parts „Quantitive analysis… …with AD compared to HC skin.” and „Since increased… …in the SC of AD skin” contain similarities with the text from reference 19.
Response:
There is no choice for completely deleting the similarity because this review article focuses on only our own published papers related to atopy pathogenesis. However, considering the reviewer’s comments about the similarities, in order to reduce the similarity as much as possible, I have extensively modified many sentences by deleting many sentences and by rewriting many sentences as marked by track changes in the revised manuscript with track changes.
------------------------------------------------------------------------------------
13th part, Accumulation of GSP:
The part „Quantitative analysis of GSP… …compared to age-matched HC skin.” contains similarities with the text from reference 16.
Page 26, at the end: SM deacylase is extensively explained, even though it was explained earlier in the text.
Word change suggestion, page 28, at the end: “Interestingly, the enzymatic reaction products, SPC and GSP, which are essential surrogates to determine whether SM GCer deacylase is functioning in situ in the epidermis, are significantly increased in the non-lesional and lesional SC of patients with AD compared with HCs, and are reciprocally related to the decreased levels of ceramides in a similar group of patients with AD.”
Page 29, Figure 25: β-glucocerebrosidase instead of glucocerebrosidase.
Response:
According to the reviewer’s suggestions. I corrected the related sentences as marked by track changes in the revised manuscript with track changes.
There is no choice for completely deleting the similarity because this review article focuses on only our own published papers related to atopy pathogenesis. However, considering the reviewer’s comments about the similarities, in order to reduce the similarity as much as possible, I have extensively modified many sentences by deleting many sentences and by rewriting many sentences as marked by track changes in the revised manuscript with track changes.
-------------------------------------------------------------------------------------
14th part, SPC Stimulates ICAM-1 Expression in Human Keratinocytes:
The part „The expression of ICAM-1… …to augment the ICAM-1 expression.” contains similarities with the text from reference 16.
15th part, SPC Stimulates TGase in Human Keratinocytes:
Similarities with reference 83.
16th part, SPC Stimulates Melanogenesis in Human Melanocytes:
Similarities with reference 91.
Response:
Considering the reviewer’ comments for the similarity, in order to completely delete the above similarities, I have completely deleted the three parts (1: SPC Stimulates ICAM-1 Expression in Human Keratinocytes, 2: SPC Stimulates TGase in Human Keratinocytes, 3: SPC Stimulates Melanogenesis in Human Melanocytes) as marked by track changes in the revised manuscript with track changes.
-------------------------------------------------------------------------------------
18th part, Purification of SM Deacylase:
Similarities with reference 106.
19th part, Enzymatic properties of Purified SM Deacylase:
Similarities with reference 106.
20th part, Identification of SM Deacylase at the Protein Level:
Similarities with reference 106.
Response:
There is no choice for completely deleting the similarity because this review article focuses only on our own published papers related to atopy pathogenesis. However, considering the reviewer’s comments about the similarities, in order to reduce the similarity as much as possible, I have extensively modified many sentences by deleting many sentences and by rewriting many sentences as marked by track changes in the revised manuscript with track changes.
-----------------------------------------------------------------------
Round 2
Reviewer 1 Report
The authors responded to all queries. The article is in my opinion publishable.
Author Response
Thank you for your comments !
Reviewer 3 Report
Brief Summary:
The manuscript has been extensively revised by the author following, wherever was possible, the previously suggested comments.
Broad comments:
Minor revisions are suggested for clarity since the performed text alterations are sometimes incomplete and not in accordance to the remaining text.
Specific Comments:
Abstract:
- The sentence “Taken together… …metabolic” looks unfinished. For clarity I would suggest: „Taken together, the sum of these findings strongly suggests that an impaired homeostasis of a ceramide-generating process may be associated with these abnormalities.”.
- The sentence “Consistently, those reaction products (SPC and GSP) accumulate to a greater extent in the involved and uninvolved SC of AD skin compared with chronic eczema or contact dermatitis skin as well as HC” is missing the word "skin" and a period "." before starting the new sentence. It should be “Consistently, those reaction products (SPC and GSP) accumulate to a greater extent in the involved and uninvolved SC of AD skin compared with chronic eczema or contact dermatitis skin as well as HC skin.”.
Discovery of Sphingomyelin Deacylase:
- For clarity, due to the deleted text, I would suggest replacing the sentence “The observed release of radiolabeled SPC from [choline-methyl-14C]SM strongly suggested that the en-zymatic reaction of SM deacylase occurs in the SC from AD skin.” with “The observed release of radiolabeled SPC from [choline-methyl-14C]SM strongly suggested the presence of a deacylase-like enzyme (SM deacylase) in the SC of AD skin.”.
SM Deacylase activity:
- For clarity in the text, I would suggest the insertion of the sentence “For the characterization of possible mechanisms involved in the ceramide deficiency of AD skin, a new quantitative assay for SM was established.” at the beginning of the paragraph.
Accumulation of GSP as Evidence for Functional GCer Deacylase:
- Word substitution of “ceramide-1” with “Cer[EOS]”.
- Addition of a comma “,” at the expression: “In comparison with SPC, there was a significant…”.
Identification of SM Deacylase at the Protein Level:
- Removal of the word “of” from the expression: “ This identification of was also corroborated…”.
- For clarity, I would suggest the substitution of the text: “aCDase is a lysosomal enzyme that catalyzes the hydrolysis of ceramides into fatty acids and SS. In the skin, aCDase is present especially in the epidermis including the SC [16, 56] and plays an essential role in producing SS which is associated with SSP-related signaling in keratinocytes [73] as well as in the ceramide-degrading process in the SC [16].” with “aCDase is a lysosomal enzyme that is present especially in the epidermis including the SC [16, 56] and catalyzes the hydrolysis of ceramides into fatty acids and SS. In the skin, SS production is associated with SSP-related signaling in keratinocytes [73] as well as in the ceramide-degrading process in the SC [16].”
- The symbol “[“ is missing in the text for reference 74.
Author Response
For Reviewer #3:
The manuscript has been extensively revised by the author following, wherever was possible, the previously suggested comments.
Broad comments:
Minor revisions are suggested for clarity since the performed text alterations are sometimes incomplete and not in accordance to the remaining text.
Specific Comments:
Abstract:
The sentence “Taken together… …metabolic” looks unfinished. For clarity I would suggest: „Taken together, the sum of these findings strongly suggests that an impaired homeostasis of a ceramide-generating process may be associated with these abnormalities.”.
The sentence “Consistently, those reaction products (SPC and GSP) accumulate to a greater extent in the involved and uninvolved SC of AD skin compared with chronic eczema or contact dermatitis skin as well as HC” is missing the word "skin" and a period "." before starting the new sentence. It should be “Consistently, those reaction products (SPC and GSP) accumulate to a greater extent in the involved and uninvolved SC of AD skin compared with chronic eczema or contact dermatitis skin as well as HC skin.”.
Response:
According to the reviewer’s kind suggestions, I have corrected the related-sentences in Abstract as marked by track changes in the re-revised manuscript with track changes.
----------------------------------------------------------------
Discovery of Sphingomyelin Deacylase:
For clarity, due to the deleted text, I would suggest replacing the sentence “The observed release of radiolabeled SPC from [choline-methyl-14C]SM strongly suggested that the en-zymatic reaction of SM deacylase occurs in the SC from AD skin.” with “The observed release of radiolabeled SPC from [choline-methyl-14C]SM strongly suggested the presence of a deacylase-like enzyme (SM deacylase) in the SC of AD skin.”.
Response:
According to the reviewer’s kind suggestions, I have corrected the related-sentences in the text as marked by track changes in the re-revised manuscript with track changes.
-------------------------------------------------------------------
SM Deacylase activity:
For clarity in the text, I would suggest the insertion of the sentence “For the characterization of possible mechanisms involved in the ceramide deficiency of AD skin, a new quantitative assay for SM was established.” at the beginning of the paragraph.
Response:
According to the reviewer’s kind suggestions, I have added the recommended-sentences in the text as marked by track changes in the re-revised manuscript with track changes.
---------------------------------------------------------------------------------
Accumulation of GSP as Evidence for Functional GCer Deacylase:
Word substitution of “ceramide-1” with “Cer[EOS]”.
Addition of a comma “,” at the expression: “In comparison with SPC, there was a significant…”.
Response:
According to the reviewer’s kind suggestions, I have replaced ceramide-1” with “Cer[EOS] and added a comma and a period in the text as marked by track changes in the re-revised manuscript with track changes.
----------------------------------------------------------------------------
Identification of SM Deacylase at the Protein Level:
Removal of the word “of” from the expression: “ This identification of was also corroborated…”.
For clarity, I would suggest the substitution of the text: “aCDase is a lysosomal enzyme that catalyzes the hydrolysis of ceramides into fatty acids and SS. In the skin, aCDase is present especially in the epidermis including the SC [16, 56] and plays an essential role in producing SS which is associated with SSP-related signaling in keratinocytes [73] as well as in the ceramide-degrading process in the SC [16].” with “aCDase is a lysosomal enzyme that is present especially in the epidermis including the SC [16, 56] and catalyzes the hydrolysis of ceramides into fatty acids and SS. In the skin, SS production is associated with SSP-related signaling in keratinocytes [73] as well as in the ceramide-degrading process in the SC [16].”
The symbol “[“ is missing in the text for reference 74.
Response:
According to the reviewer’s kind suggestions, I have removed “of” and replaced the indicated sentences with the recommended sentences in the text as marked by track changes in the re-revised manuscript with track changes.